# Dynamic interplay of microtubule and actomyosin forces drive tissue extension

Amrita Singh [1,2,5], Sameedha Thale [1,2,5], Tobias Leibner[3], Lucas Lamparter[2,4], Andrea Ricker[4], Harald Nüsse[4], Jürgen Klingauf[2,4], Milos Galic [2,4], Mario Ohlberger [3] & Maja Matis [1,2,4] ✉

In order to shape a tissue, individual cell-based mechanical forces have to be integrated into a global force pattern. Over the last decades, the importance of actomyosin contractile arrays, which are the key constituents of various morphogenetic processes, has been established for many tissues. Recent studies have demonstrated that the microtubule cytoskeleton mediates folding and elongation of the epithelial sheet during *Drosophila* morphogenesis, placing microtubule mechanics *on par* with actin-based processes. While these studies establish the importance of both cytoskeletal systems during cell and tissue rearrangements, a mechanistic understanding of their functional hierarchy is currently missing. Here, we dissect the individual roles of these two key generators of mechanical forces during epithelium elongation in the developing *Drosophila* wing. We show that wing extension, which entails columnar-to-cuboidal cell shape remodeling in a cell-autonomous manner, is driven by anisotropic cell expansion caused by the remodeling of the microtubule cytoskeleton from apico-basal to planarly polarized. Importantly, cell and tissue elongation is not associated with Myosin activity. Instead, Myosin II exhibits a homeostatic role, as actomyosin contraction balances polarized microtubule-based forces to determine the final cell shape. Using a reductionist model, we confirm that pairing microtubule and actomyosin-based forces is sufficient to recapitulate cell elongation and the final cell shape. These results support a hierarchical mechanism whereby microtubule-based forces in some epithelial systems prime actomyosin-generated forces.

Tissue morphogenesis relies on a finely tuned spatial and temporal integration of various cell behaviors, including changes in cell shape, size, migration, division, and intercalation[1,2]. These distinct behaviors are driven by tissue intrinsic and extrinsic mechanisms, which jointly coordinate mechanical forces exerted on cells during tissue patterning[3–5]. Within individual cells, actomyosin filaments, together with microtubules and intermediate filaments, form the composite cytoskeleton that controls cell mechanics during tissue remodeling. While studies established the importance of actin-based mechanical forces[6], relatively little is known about the contribution of other cytoskeletal components to cell shape changes and cell mechanics during morphogenesis.

Microtubules were initially considered to participate only in a supporting role, contributing, for instance, to the stabilization of

[1]Institute of Cell Biology, Medical Faculty, University of Münster, Münster, Germany. [2]Cells in Motion' Interfaculty Centre, University of Münster, Münster, Germany. [3]Applied Mathematics, Institute for Analysis and Numerics, Faculty of Mathematics and Computer science, University of Münster, Münster, Germany. [4]Institute of Medical Physics and Biophysics, Medical Faculty, University of Münster, Münster, Germany. [5]These authors contributed equally: Amrita Singh, Sameedha Thale. ✉e-mail: matism@uni-muenster.de

actomyosin or trafficking of adhesion molecules[7–9]. However, recent work has challenged this view, demonstrating that microtubules are capable of generating forces that are crucial for key morphogenetic processes, including tissue bending and tissue extension[10–13]. Microtubules can directly generate protrusive and pulling forces through polymerization and depolymerization, respectively, or together with motor proteins[13]. Importantly, however, not only the active generation of forces but also changes in the mechanical properties of microtubules play a role in cell mechanics[14,15]. Yet, while these studies demonstrate that actomyosin and microtubule-based forces are equally important, how these force-generating systems interact in space and time remains unclear.

Early *Drosophila melanogaster* wing development presents a versatile in-vivo model system to unravel the hierarchy between actin and microtubule-based forces. The developing wing epithelium undergoes dynamic 3D changes, which can be divided into three phases: initial adhesion, separation, and reapposition[16–18]. In the first phase, the single-layered wing disc epithelium undergoes eversion, resulting in the adhesion (i.e., apposition) of newly formed dorsal and ventral surfaces at their basal sites (phase I). Approximately 8 h after pupal formation (APF), both epithelial layers separate, elongate, and reapposition (phase II). In the last phase, at approximately 18 hAPF, the hinge of the wing starts to contract, resulting in the final wing shape at 30 hAPF.

The main mechanism driving global tissue remodeling are shape changes at the cellular level. In the first phase, cells transition from a columnar to a cuboidal shape, which concomitantly reduces the apico-basal cell height and increases cell width, resulting in wing tissue flattening and expansion[19]. In the second phase, cells first increase in lateral height (approximately 12 hAPF) and then elongate along the proximal-distal (P/D) axis[11,20,21]. After reapposition of the dorsal and ventral surfaces at 18 hAPF, the wing hinge contraction leads to cell rearrangements, generating a final hexagonal cell pattern at 30 hAPF[21–23]. While previous studies have established the presence of microtubule and Myosin II-generated forces in developing wing tissue[11,24], the exact contribution of these systems to cell and wing shape remains an open and intriguing question. In this study, we explored the interplay of actin and microtubule-based forces during tissue elongation along the P/D axis. Combining quantitative cell biological and numerical approaches, we show that the dynamic reorganization of microtubule-generated forces within the plane drives cell elongation and that the subsequent steady-state distribution of microtubules and actomyosin forces results in cell length homeostasis during epithelium development.

## Results
### Initial tissue remodeling occurs by columnar-to-cuboidal cell shape change in a cell-autonomous manner
To properly investigate what mechanical forces contribute to cell shape changes during wing morphogenesis requires a clear definition of tissue dynamics. We started to analyze the wing at 14.5 hAPF after the head eversion when both layers reached maximal separation[16,25]. Quantification of wing tissue changes between 14.5 and 18 hAPF showed elongation of the wing along the P/D axis (Fig. 1a–c; Supplementary Fig. 2a). Imaging of GFP-tagged Disc-large (Dlg), which marks the basolateral membrane, revealed that tissue elongation within the plane was accompanied by columnar-to-cuboidal cell shape changes (Fig. 1d–f). The observed shape transition, which reduces cell height and increases cell width, is one of the main mechanisms driving epithelial tissue expansion during morphogenesis[2]. However, it does not explain the observed anisotropic growth of the wing tissue. Therefore, we quantified cell shape changes within the plane between 14.5 and 18 hAPF. We found that the cell apical area changes anisotropically by elongating along the P/D axis (Fig. 1h, i; Supplementary Fig. 1a–c), suggesting that the elongation of cells within the plane drives tissue

elongation along the same axis. Importantly, the cell area is not strongly changing over time, establishing that the observed increase in cell length along the P/D axis is caused by elongation (Fig. 1g). Although we observed the continuous elongation of cells, we observed the largest change during the initial elongation of the wing between 14.5 and 16 hAPF, followed by minor changes between 16 and 18 hAPF.

We then asked if tissue elongation is driven by cell-intrinsic or cell-extrinsic forces. Cell intrinsic forces, which are exerted on the cell cortex, are primarily the result of the cytoskeleton. On the other hand, tensile and compressive forces applied to cells from external loads can also drive the remodeling of tissues. Indeed, the initial step in wing development during the first apposition depends on local forces[19], while wing tissue remodeling during the last phase depends on extrinsic tensile forces generated by wing hinge contraction that starts at 18 hAPF[21,24,26,27]. During wing hinge contraction, the apical extracellular matrix protein Dumpy anchors the wing epithelium to the adjacent cuticle, resulting in tension along the P/D axis[26,27]. To account for a possible cell-extrinsic contribution to initial cell and tissue elongation (Fig. 1a), we explored potential extrinsic mechanical forces before 18 hAPF by three independent approaches. First, we verified *dumpy* mutants, in which the anchorage of the epithelium to the cuticle is perturbed. In the *dumpy* mutant, the tissue elongated normally compared to the control, thus confirming that hinge contraction does not contribute to elongation until 18 hAPF (Fig. 1a–c). To further disentangle active cell forces from forces imposed extrinsically on wing cells, we next severed the wing blade from the hinge at 18 hAPF (i.e., after the initial cell elongation and just before the onset of hinge contraction) and quantified the cell length (Supplementary Fig. 2c; Supplementary Movie 1). We reasoned that if the initial cell elongation is independent of cell-extrinsic forces, cells should not shorten upon uncoupling of the blade from the hinge. Consistently, the analysis revealed that cells stayed elongated after 1 h (Supplementary Fig. 2b). To further validate these findings and exclude the stiff apical cuticle material as the possible mechanical influence that may cause cells to stay elongated after being mechanically isolated, we performed TEM analysis of 16 hAPF old wings. We unequivocally observed that the epithelium was molted at this stage (Supplementary Fig. 2d), thus excluding a contribution of the apical cuticle to cell shape changes. The measured changes in tissue reveal that morphogenetic elongation of the wing occurs by columnar-to-cuboidal cell shape change in a cell-autonomous manner. Therefore, we focused on local cell intrinsic forces for the remainder of the study.

### Cell elongation is associated with the remodeling of the microtubule cytoskeleton
Cell shape changes can be driven by the dynamic remodeling of cell-cell adhesion, by forces generated by contractile actomyosin cytoskeleton, or by the polarization of actin and microtubules at cellular interfaces. To better characterize wing cell and tissue elongation, we first quantified the evolution of the main regulators of cell shape changes between 14.5-18 hAPF: E-cadherin (E-cad), which mediates cell adhesion, as well as actomyosin and microtubules, which generate intracellular forces.

Notably, we did not observe changes in E-cad levels at the junctions during tissue elongation (Fig. 2a, b) or distribution (Fig. 2c). Moreover, we did not find changes in the levels or localization of non-muscle Myosin II levels (by visualizing the Myosin Regulatory Light Chain, Sqh) which argues against the role of Myosin II at the onset of cell elongation (Fig. 2d–f). Consistent with the previous observations[11], we only observed changes in Myosin II at 18 hAPF when the levels of Myosin II increased and became planarly polarized along the longer P/D oriented junctions (Fig. 2e, f). Next, we visualized the microtubule cytoskeleton between 14.5-18 hAPF (i.e., the columnar-to-cuboidal cell shape transition). Notably, we found extensive reorganization of microtubules during initial cell elongation (Fig. 2g). Live imaging of

 

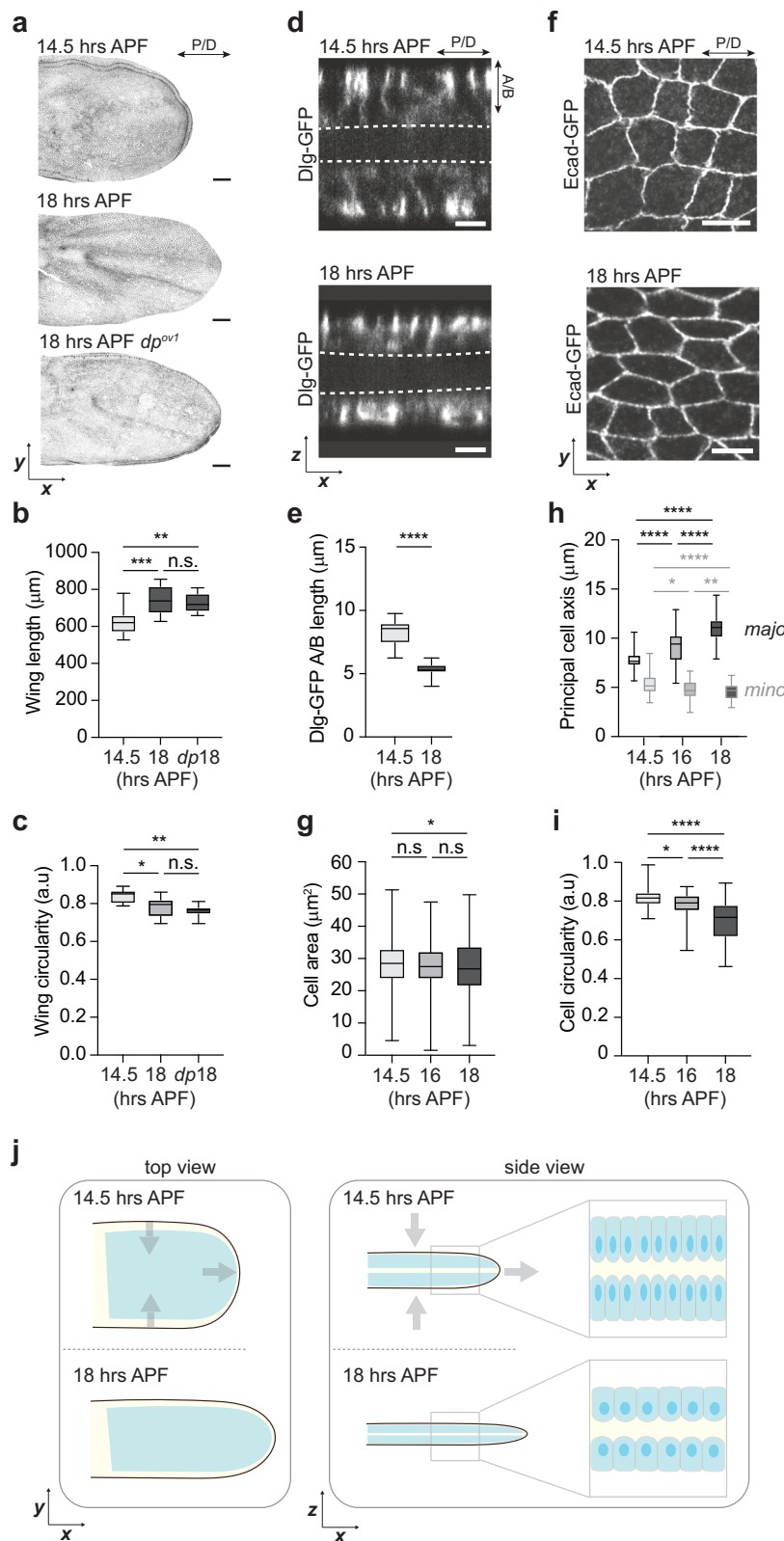

EOS-α-tubulin (EOS-Tub) showed that at 14 hAPF, when cells were still columnar, and their apical surface displayed a round shape (Fig. 1d, f), microtubules formed lateral bundles organized along the apico-basal axis (Fig. 2g, Supplementary Movie 2). This is consistent with a previous study showing the presence of lateral microtubule bundles in wing epithelium during head eversion[20]. Notably, within the next 30 minutes, the microtubule cytoskeleton drastically reorganized,

resulting in the appearance of a dense apical planar microtubule network (Fig. 2j). Specifically, EOS-Tub showed that newly established planar microtubules aligned along the P/D axis (Fig. 2k). The formation of a planar microtubule network precedes the elongation of cells along the same axis (Fig. 1). At the same time, the lateral microtubule bundles disassembled, and the EOS-Tub signal in the basolateral part of the cell is not visible at 16 hAPF (Fig. 2g).

**Fig. 1 | Wing elongation occurs by columnar-to-cuboidal cell shape change in a cell-autonomous manner. a** Representative images of developing 14.5 (top) and 18 hAPF (middle) control (*w*[*1118*]) and 18 hAPF *dumpy* (bottom) wing tissue. During pupal wing morphogenesis, the epithelium undergoes elongation independent of extrinsic forces, as the wing normally elongates in the *dumpy* mutant. **b** Graph showing quantification of wing length for developing control and 18 hAPF *dumpy* mutant wings as indicated by the red line in Supplementary Fig. 2a (Ordinary one-way ANOVA: **$p = 0.0061$, ***$p = 0.0007$ and n.s. $p = 0.7897$, $N$ (wings) = 10, 14, 11). **c** Quantifications of circularity for control and 18h APF *dumpy* mutant wing tissue (Ordinary one-way ANOVA: **$p = 0.0037$, *$p = 0.0422$ and n.s. $p = 0.4564$, $N$ (wings) =10, 14, 11). **d** Orthogonal views of 14.5 and 18 hAPF wings marked by Dlg-GFP to visualize the basolateral membrane (top). **e** Quantifications of Dlg-GFP along the apical-basal (A/B) axis for 14.5 and 18 hAPF control wings show that wing elongation occurs by columnar-to-cuboidal cell shape change (two-tailed t-test, ****$p < 0.0001$, $n(15) = 34$ and $n(18) = 29$ cells and 4 pupae per genotype). **f** Images indicating the cell apical side at 14.5 and 18 hAPF were marked by E-cad-GFP to visualize the cell shape. **g** Quantifications of cell area for developing (14.5-18 hAPF) control (*w*[*1118*]) cells (Kruskal–Wallis test: *$p = 0.0198$, n.s. $p = 0.1690$, n.s. $p > 0.999$, $n(14.5) = 287$, $n(16) = 298$ and $n(18) = 302$ cells and 3 pupae). **h** Quantification of the cell major axis (along the P/D axis) and minor axis (along A/P axis) for developing wings at 14.5, 16, and 18 hAPF (Kruskal–Wallis test for major axis: ****$p < 0.0001$, ****$p < 0.0001$ and ****$p < 0.0001$ and minor axis: ****$p < 0.0001$, *$p = 0.0461$ and **$p = 0.0096$). **i** Quantifications of circularity for developing (14.5–18 hAPF) control (*w*[*1118*]) cells (Kruskal–Wallis test: ****$p < 0.0001$, *$p = 0.0368$ and ****$p < 0.0001$). For (**h**, **i**), $n(14.5) = 97$, $n(16) = 93$ and $n(18) = 112$ cells and 3. Boxes in all plots extend from the 25th to 75th percentiles, with a line at the median. Whiskers show min and max values. **j** Schematic representation of columnar-to-cuboidal cell shape changes with observed shortening of cell height and anisotropic increase in cell width, resulting in wing tissue elongation. Source data are provided as a Source Data file. Scale bars, **a** 50 μm and (**d**, **f**) 5 mm.

Altogether, we found that cell elongation is not associated with significant changes in E-cad or Myosin II levels. We could only observe modest but significant accumulation and polarization of the Myosin II after cells had already undergone elongation, decoupling Myosin II activity from this process. Our observation that cell shape changes are concurrent with the remodeling of the microtubule cytoskeleton suggests a direct role of microtubules in controlling cell shape changes.

### Actomyosin contractility is not required for initial cell elongation but contributes to cell shape refinement

Since actomyosin contractile forces are the primary driver of various forms of cell rearrangements during epithelium remodeling[28], we wanted to confirm that the observed late changes in Myosin II are dispensable for cell elongation. To that end, we took advantage of a degron-based protein knockdown system (deGradFP) to selectively disrupt Myosin II function[29]. The deGradFP system was efficiently used to deplete the GFP-tagged Myosin II regulatory light chain (Sqh-GFP) at different developmental stages[29,30]. Since expression of deGradFP resulted in early embryonic lethality of males, we used the tub-Gal80[ts] system to express it only in the pupal stage (Supplementary Table 1, Supplementary information). The expression of the deGradFP system disrupted actomyosin cables in Myosin II-depleted cells (Supplementary Fig. 3a). We next performed laser ablation experiments to test for changes in tension at adherens junctions along the P/D axis (Supplementary Fig. 3b). Compared to controls (Sqh-GFP males and deGradFP females), deGradFP males displayed a strong drop in recoil velocity (Supplementary Fig. 3d), indicative of reduced tension along the Myosin II enriched junctions. Consistently, deGradFP-mediated knockdown strongly reduced overall Myosin II phosphorylation (Supplementary Fig. 3c, e). Notably, deGradFP-mediated knockdown of Sqh-GFP presented phenotypes reminiscent of Myosin II loss-of-function mutants in *Drosophila* (Supplementary Fig. 4). Finally, deGradFP females, which carry a wild-type copy of Myosin II, show no defects observed in deGradFP males providing evidence that the system does not cause any Myosin II-unspecific phenotypes (Supplementary Fig. 3f, g). Next, we analyzed cell shape in Myosin II-depleted cells. Quantification of cell length showed that cells depleted of Myosin II were significantly longer than control cells (Fig. 3a, b).

Having established that Myosin II is dispensable for initial cell elongation, we looked closer at its role in other aspects of cell shape changes. Myosin II appeared to be required for the final cell shape changes when its junctional level increased and became polarized (Fig. 2d). This was evident from the change in curvature of cellular interfaces in deGradFP males, where Myosin II contractility was disrupted, and cortical tension was released. In addition, cells in Myosin II-depleted wings expanded and widened along the anterior-posterior (A/P) axis (Fig. 3c), similar to 16 hAPF control cells (Fig. 1h). Hence, our results demonstrate that polarized Myosin II contractility is not required for the initial polarized cell elongation along the P/D axis but for subsequent refinement of cell shape (Fig. 3d, g). In addition, it posits that Myosin-independent forces are needed for cell elongation.

### Microtubules initiate polarized cell elongation along the P/D axis

As shown above, wing cell elongation entails microtubule remodeling from apico-basal to planar polarized alignment along the P/D axis (Supplementary Movies 2, 3). We thus hypothesized that in Myosin II-depleted cells, where cortical tension is reduced, microtubule-generated protrusive forces might drive further cell elongation. Should this indeed be the case, polarized microtubule patterning should also occur in deGradFP flies lacking actomyosin contractility. However, microtubule alignment along the P/D axis was comparable for deGradFP and control flies (Fig. 3e). Quantifying the angular distribution of microtubules with respect to the P/D axis showed no significant difference between Myosin II depleted and control cells (Fig. 3f). These data indicate that the patterned microtubule cytoskeleton directs cell shape changes.

To further strengthen the validity of this hypothesis, we probed whether the misalignment of microtubules results in shorter cells. Microtubule patterning in wing cells is regulated through the Fat-PCP signaling pathway[11,31–33]. Notably, the *ft-PCP* mutant animals rescued for the Hippo pathway (hereafter called *ft-PCP* mutant) in which microtubules are misoriented (Fig. 4a, b) display a smaller cell elongation index (EI, defined by the ratio of the length of the longest cell axis to the shortest cell axis)[11]. Considering that an increase in cell width and a decrease in cell length both lower the EI, we next measured the absolute cell length in the *ft-PCP* mutant. Consistent with the role of planar polarized microtubules for cell elongation, we found that cells are shorter in the *ft-PCP* mutant background (Fig. 4d–f).

We reasoned that in *ft-PCP* mutant cells, microtubules may not undergo correct remodeling from lateral to planar polarization, and thus, cells fail to elongate. Consistent with the hypothesis, the organization of microtubules in the *ft-PCP* mutant resembles an epithelial-like microtubule array spread along the apico-basal axis at the 18 hAPF (Supplementary Fig. 5a, b, Supplementary Movie 4). We verified these data using transmission electron microscopy (TEM), which shows most of the microtubules aligned along the apico-basal axis (Supplementary Fig. 5c, d). Importantly, microtubules before the columnar-to-cuboidal cell transition are organized in lateral bundles similar to those in wild-type cells (Supplementary Fig. 5e). This is consistent with previous report showing that the *ft-PCP* mutant animals rescued for the Hippo pathway do not display defects in tissue shape at earlier stages[34].

Considering that disrupted cell polarity in the *ft-PCP* mutant may lead to cell shape changes independent of microtubule-generating

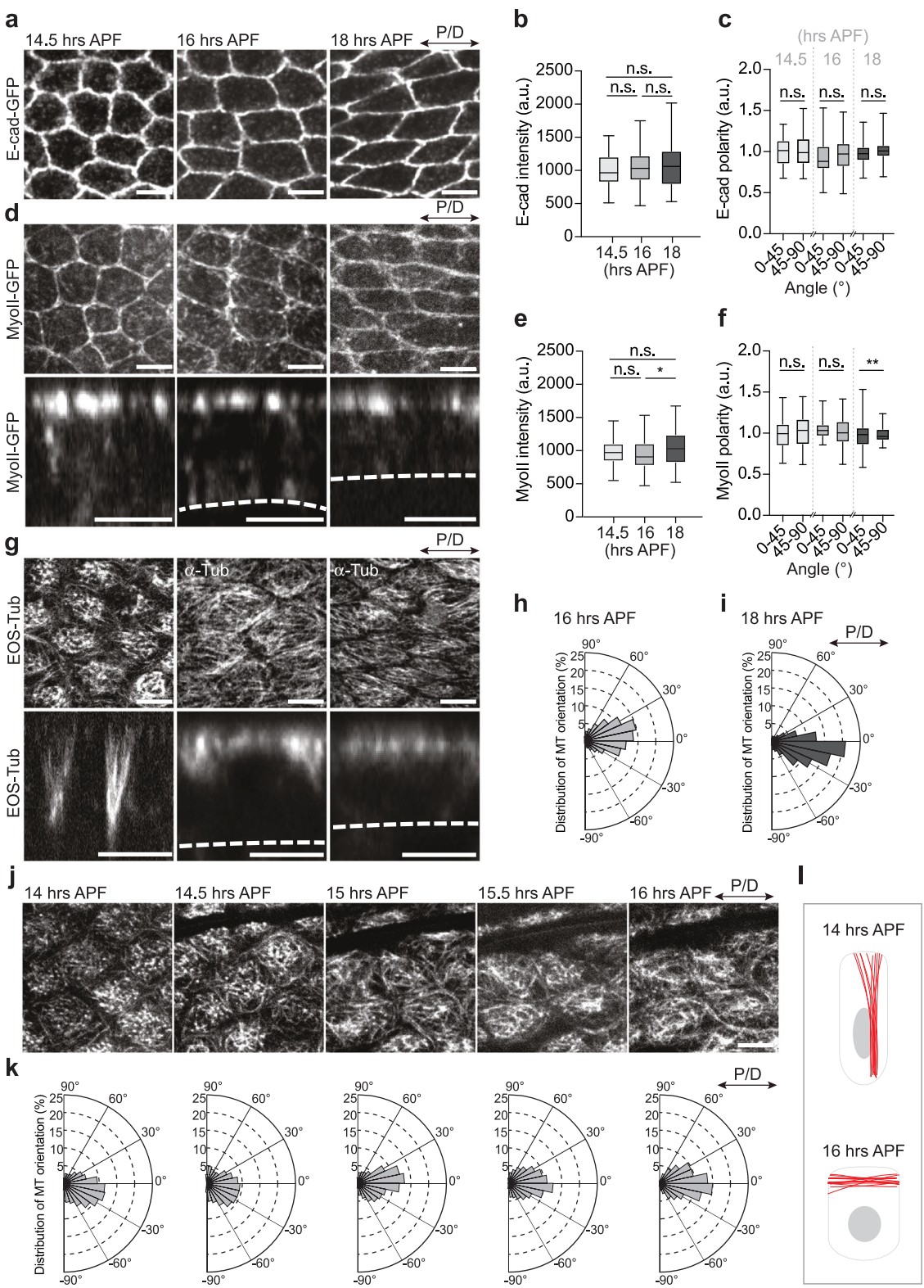

forces, we used Patronin to specifically perturb microtubule organization. Members of the calmodulin-regulated spectrin-associated protein (CAMSAP) family in vertebrates and Patronin in invertebrates play an essential role in organizing the microtubule cytoskeleton in several differentiated cells[12,35–38]. Consistently, Patronin knockdown in wing cells resulted in a substantial change in the organization of apical non-centrosomal microtubules (Fig. 4a–c, Supplementary Movie 5). Importantly, Patronin-depleted cells were also shorter (Fig. 4d–g).

Finally, considering that microtubules drive cell elongation along the P/D axis, we reasoned that stabilization of microtubules may be important. To probe this possibility, we overexpressed Patronin in the wing cells (Fig. 4h). Consistent with our hypothesis, cells were longer, while their apical area stayed constant (Fig. 4i–l). Collectively, these independent experimental approaches consistently demonstrate that the dynamic remodeling of microtubules within the tissue plane drives cell elongation and that the subsequent balance between microtubule and actomyosin forces results in cell-length homeostasis.

**Fig. 2 | Cell elongation is associated with remodeling of the microtubule cytoskeleton. a** Images of wings expressing E-cad-GFP at 14.5, 16, and 18 hAPF. **b** Quantifications of mean intensities of E-cad for 14.5 (light gray), 16 (gray), and 18 (dark gray) hAPF (Kruskal–Wallis test: n.s. = 0.1465, n.s. = 0.6310 and n.s. > 0.9999. **c** Quantifications of E-cad polarity in 14.5, 16, and 18 hAPF control cells (Kruskal–Wallis test: n.s. > 0.9999, n.s. = 0.3490 and n.s. = 0.4872, *n*(14.5) = 64/54, *n*(16) = 73/52 and *n*(18) = 100/60 junctions and 3-5 pupae per genotype). **d** Images of wings expressing Sqh-GFP (14.5, 16, and 18 hAPF) to visualize Myosin II. The lower panel indicates the orthogonal views with a white dashed line marking the basal side of the cells. **e** Quantifications of mean intensities of Myosin II for 14.5 (light gray), 16 (gray), and 18 (dark gray) hAPF (Ordinary one-way ANOVA: n.s.= 0.1455, n.s. = 0.2693 and * *p* = 0.0164, *n*(14.5) = 128, *n*(16) = 106 and *n*(18) = 123 cells and 3-5 pupae per genotype). **f** Quantification of myosin II polarity in 14.5, 16, and 18 hAPF control cells (Kruskal–Wallis test: n.s. > 0.9999, n.s. = 0.4474 and ** *p* = 0.0091, *n*(14.5) = 79/49, *n*(16) = 64/42 and *n*(18) = 90/57 junctions and 3-5 pupae per genotype). For (**b, c, e, f**), boxes in the plot extend from the 25th to 75th percentiles, with a line at the median. Whiskers show min and max values. **g** Images of control cells (14.5, 16, and

18 hAPF) marked by EOS-Tub or α-Tub antibody to visualize microtubules. The lower panel indicates the orthogonal views with a white dashed line marking the basal side of the cells. The images shown are representative of 4 wings and 3 independent experiments. **h, i** Distribution of microtubule orientation along the P/D axis for 16 and 18 hAPF control wings. Microtubule orientation pooled together and binned into 3 categories of angular distribution −30° to 0° and 0° to 30° (along the P/D axis), −60° to −30° and 30° to 60°, −90° to −60° and 60° to 90° for comparison along different axes at 16 and 18 hAPF (Kruskal-Wallis test: **** *p* < 0.0001). **j** Live imaging of developing wing epithelium (14, 14.5, 15, 15.5, and 16 hAPF) marked by EOS-Tub indicating remodeling of microtubules in the apical side along the P/D axis of the wing. The images shown are representative of 4 wings and 3 independent experiments. **k** Distribution of microtubule orientation along the P/D axis between 14 and 16 hAPF control cells for images in (**j**). **l** Cartoon showing a cross-section of wing cells at 14 and 16 hAPF depicting the organization of the microtubule cytoskeleton in developing tissue. Source data are provided as a Source Data file. Scale bars, (**a, j**) 5 μm, (**d, g**) 5 μm.

## Cell-based forces shape wing morphology

The findings to this point argue for an interplay of microtubule and actomyosin forces as a regulatory mechanism driving cell shape changes during tissue morphogenesis. To validate this observation, we systematically tested the role of microtubule and actin-based forces during wing elongation. The loss of Myosin II function strongly affected the tissue shape, arguing that changes at the cellular level translate into tissue-level changes. The anterior compartment, where Myosin II function was inhibited, was significantly longer than the posterior compartment used as an internal wild-type control (Fig. 5a, b). This is in line with our result showing that cortical tension inversely correlates with cell elongation (Fig. 1h; Supplementary Fig. 3b).

Given that the Fat-PCP signaling pathway is required to direct microtubule-generated forces, we probed for changes in the length of the *ft-PCP* mutant wing epithelium. In agreement with our findings at the cellular level, perturbation of the microtubule cytoskeleton led to a significant defect in tissue elongation (Fig. 5c–e, Supplementary Fig. 6a, b). Moreover, *ft-PCP* mutant tissue is similar in size and shape to the wild-type wing at the onset of cell elongation (Figs. 5d, 1b), arguing that Fat-PCP-dependent planar patterning of forces is required for cell and tissue shape. Similarly, the analysis of Patronin-depleted wings showed a significant reduction in pupal and adult wing length, thus confirming the specific requirements of microtubules for cell and tissue elongation (Fig. 5c–e; Supplementary Fig. 6a, b). Consistently, the hinge contracts normally in *ft-PCP* mutant and Patronin-depleted wings (Supplementary Fig. 6c), confirming that the tissue elongates independent of cell-extrinsic forces generated by hinge constriction.

Finally, to assess the relationship between microtubule-generated forces and tissue elongation and to rule out unspecific effects resulting from possible interaction between the Fat-PCP pathway and Myosin II, we probed whether the global level of phosphorylated Myosin II and its polarity is preserved in *ft-PCP* mutant tissue. This was the case despite changes in cell shape (Supplementary Fig. 5f–i), revealing that the Fat-PCP signaling pathway acts upstream of the microtubule cytoskeleton in the wing epithelium but does not regulate actomyosin.

Taken together, our analysis of tissue morphogenesis illustrates that spatial patterning of local forces generated by microtubules and not by actomyosin contractile forces is essential for tissue extension. We propose that gradients of Ds and Fj, constituents of the Fat-PCP signaling pathway, serve as instructive cues at the cell and tissue levels to pattern forces.

## A model of force balance predicts polarized microtubule-generated forces as an essential driver of cell elongation

We aimed to formalize these findings in a force-balance model to test whether microtubule-generated forces counterbalanced by contractile actomyosin forces are sufficient to drive observed cell elongation.

Various continuum and agent/vertex-based models have been devised on the cell level and validated in different contexts[39]. Furthermore, attempts have been undertaken to rigorously derive macroscopic continuum models on the tissue level, starting from individual cell-based models[40]. In most of these settings, however, the effects of the microtubule or cytoskeleton reorganization are not taken into account. Therefore, we developed and implemented a continuum model that is adapted to our setting. Since, in our case, the cells are mechanically autonomous[11], it suffices to model a single cell. To average out microscopic details while maintaining the structural properties, the model regards the cell from a mesoscopic point of view (Fig. 6a). The microtubule cytoskeleton is modeled as a viscoelastic gel formed by polar filaments that can actively exert forces (active polar gel)[41]. The actomyosin cortex is modeled through the effective tension of the cell surface. For full details on the mathematical model and its efficient numerical discretization and implementation, see ([42] and Materials and Methods section).

As shown in Fig. 6, our model recapitulates the overall cell shape changes observed in vivo. For the right choice of parameters, a force balance is obtained such that the cell approximately maintains its shape in a steady state (Fig. 6b, c). If the microtubule-based forces are reduced, modeling a disassembly or misorientation of the microtubules, the cell shortens along the P/D axis (Fig. 6d, compared to Fig. 4a, d). On the other hand, for reduced actomyosin forces (reduced effective surface tension), the cell becomes significantly elongated (Fig. 6e, compared to Fig. 3a, b).

## Discussion

In this study, we delineate the roles of Myosin II and microtubules in wing cell elongation. We found that the rapid remodeling of the microtubule cytoskeleton, from apico-basal to planarly polarized, is concomitant with the onset of the columnar-to-cuboidal cell shape transition. Our observations can be quantitatively explained by the collective action of dynamic, oriented microtubules. The formation of the apical planar network, which depends on the Ft-PCP signaling pathway, yields a non-isotropic force distribution that is required for cell and wing tissue extension. While our findings are consistent with microtubule-based protrusive forces, as observed in vitro[13], the lack of direct force measurements in our model system precludes this conclusion. Notably, we observed no role for actomyosin in cell elongation. We found instead that actomyosin contractility refines the final cell shape by limiting cell length and narrowing these cells. Collectively, our work fills a critical gap in understanding the interplay between local microtubule and actomyosin-generated forces during wing elongation. Furthermore, as depletion of Myosin II leads to the isotropic junctional tension and cell prestress release, it argues against the presence of additional polarized contractile force in wing cells.

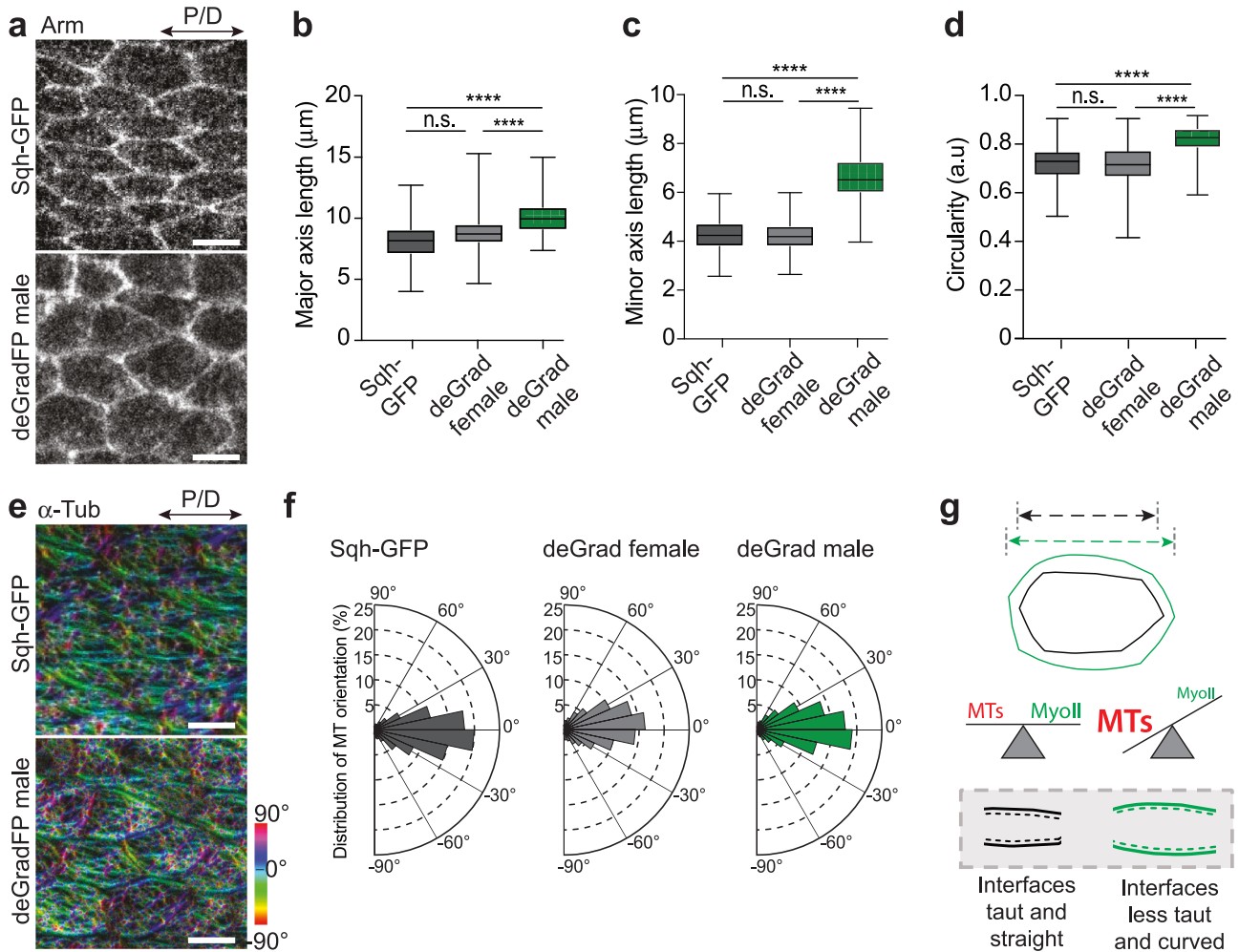

**Fig. 3 | Actomyosin contractility is not required for initial cell elongation but contributes to cell shape refinement. a** Images of 18 hAPF wings of Sqh-GFP male (*sqh^AX3/Y; sqh-Sqh-GFP/tubGal80^ts*; ciGal4/+) and deGradFP male (*sqh^AX3/Y; sqh-Sqh-GFP/tubGal80^ts*; ciGal4/UAS-NSlmb-vhhGFP4) stained for Arm to visualize cell shape. **b** Quantifications of the cell major axis (along the P/D axis) for different genotypes (Kruskal–Wallis test, n.s. *p* > 0.9999, **** *p* < 0.0001, **** *p* < 0.0001). **c** Quantifications of the cell minor axis (along the A/P axis) for different genotypes (Kruskal–Wallis test from left to right: **** *p* < 0.0001, n.s. *p* > 0.9999, **** *p* < 0.0001). **d** Quantifications of cell circularity for different genotypes (Kruskal–Wallis test from left to right: **** *p* < 0.0001, n.s. *p* = 0.6690, **** *p* < 0.0001). For (**b**–**d**) n(SqhGFP male) = 250 cells, n(deGradFP (*sqh^AX3/+; sqh-Sqh-GFP/tubGal80^ts*; ciGal4/UAS-NSlmb-vhhGFP4) female) = 250 and n(deGradFP male) = 250 and 3–5 pupae per genotype. Boxes in the plot extend from the 25th to the 75th percentiles, with a line at the median. Whiskers show min and max values. **e** Images of 18 hAPF wings of Sqh-GFP male and deGradFP male stained for α-Tub to visualize microtubules. The orientation of microtubules is color-coded using OrientationJ. The images shown are representative of 4 wings and 3 independent experiments. **f** Distribution of microtubule orientation along the P/D axis for Sqh-GFP male, deGradFP female and deGradFP male. Microtubule orientation pooled together and binned into 3 categories of angular distribution −30° to 0° and 0° to 30° (along the P/D axis), −60° to −30° and 30° to 60°, −90° to −60° and 60° to 90° for comparison across genotypes (Kruskal-Wallis test: n.s. *p* > 0.9999 for each bin of Sqh-GFP male compared to deGradFP female and deGradFP male). **g** Cartoon showing the effect of disruption of Myosin II and microtubule activity on cell shape. The cell shape changes result from a general release of cellular prestress upon loss of Myosin II contractility, as suggested by an increase in length and width of the cell. Thus, there is no direct relation between Myosin II polarized organization and cell elongation along the P/D axis. Importantly, the reduction in Myosin II contractility may lead to tissue softening, which may also contribute to the observed cell shape changes. However, Myosin II refines the overall cell shapes by (i) regulating the length of the cells and (ii) keeping the interfaces along the P/D axis taut and straight (indicated by black broken lines), as shown through the region of cell interfaces marked within the gray box (**g**, bottom). Source data are provided as a Source Data file. Scale bars, (**a**, **e**) 5 μm.

Consistently, the downregulation of Dachs and MyosinVI, two atypical myosin motor proteins previously associated with junctional remodeling[43,44], did not affect wing shape (Supplementary Fig. 8).

We further find that the patterning of non-centrosomal microtubules is independent of mechanical or geometrical cues. Cell shape was proposed to play a critical role in aligning non-centrosomal microtubules in the fly epithelium[45]. Considering their stiffness and angular dependency on catastrophe, growing microtubules could self-organize into a network controlled by the elongated cell shape[46,47]. However, the patterning of microtubules within the plane, which is required for cell elongation, precedes columnar-to-cuboidal cell shape changes (Fig. 2j). Hence, our data

support the notion that cell elongation is not the cause but the consequence of the polarized alignment of microtubules via the Fat-PCP signaling pathway. Indeed, failure to pattern apical microtubule cytoskeleton upon loss of the microtubule minus-end binding protein Patronin, which is required for the correct organization of non-centrosomal microtubules in wing epithelial cells, results in shorter cells and tissue.

In summary, our results refine the current view on wing epithelium development, showing that a combination of local opposing forces drives cell elongation. Importantly, the observed mechanical autonomy of cells at this early stage is consistent with microtubule perturbation experiments, where cells fail to elongate (Fig. 4d). In both

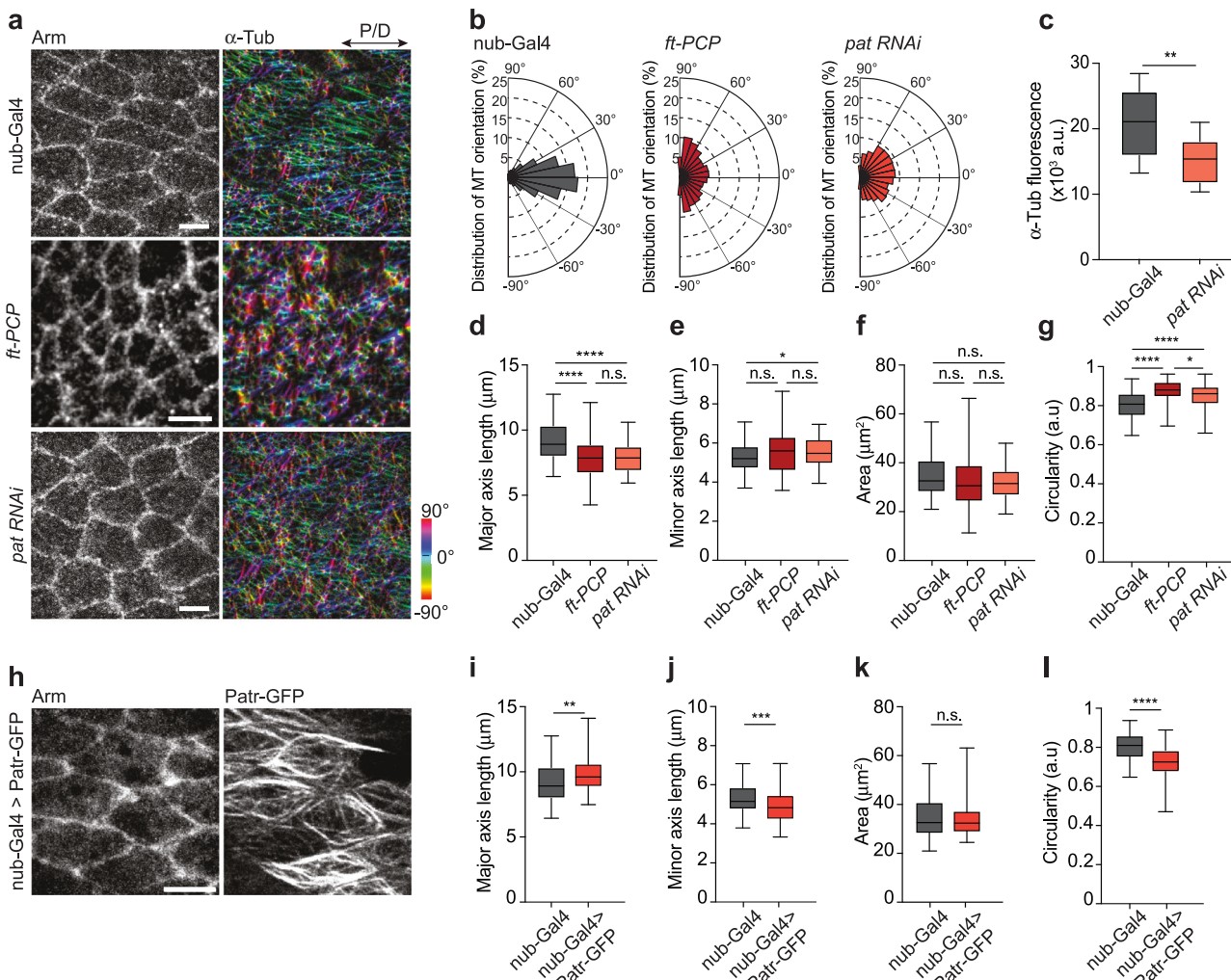

**Fig. 4 | Remodeling of microtubules initiates polarized cell elongation along the P/D axis. a** Representative images of 18 hAPF control (*nub-Gal4*), *ft-PCP* (*ft^(2),fd^/ft^GRV;act-Gal4/UAS-FtΔECDΔN-1*) and Patronin-depleted (*nub-Gal4>Patronin^RNAi*) stained for Arm and α-Tub to visualize microtubules. The orientation of microtubules is color-coded using OrientationJ. The images shown are representative of 4 wings and 3 independent experiments. **b** Distribution of microtubule orientation along the P/D axis for control, *ft-PCP*, and Patronin-depleted wings. Microtubule orientation pooled together and binned into 3 categories of angular distribution −30° to 0° and 0° to 30° (along the P/D axis), −60° to −30° and 30° to 60°, −90° to −60° and 60° to 90° for comparison across genotypes (Kruskal-Wallis test: **** $p < 0.0001$ for each bin of control compared to *ft-PCP* mutants and Patronin-depleted wings). **c** Quantification of microtubule intensities in control and Patronin-depleted wings (two-tailed t-test: ** $p = 0.0017$, $n$(*nub-Gal4*) = 96 and $n$(*nub-Gal4>Patronin^RNAi*) = 122 cells.) **d** Quantifications of the cell major axis (along the P/D axis) for different genotypes (Kruskal–Wallis test from left to right: **** $p < 0.0001$, **** $p < 0.0001$ and n.s. $p > 0.9999$). **e** Quantifications of the cell minor axis (along the A/P axis) for different genotypes (Kruskal–Wallis test from left to right: * $p = 0.0221$, n.s. $p = 0.0799$ and n.s. $p > 0.9999$). **f** Quantifications of cell

area for different genotypes (Kruskal–Wallis test from left to right: n.s. $p = 0.1240$, n.s. $p = 0.2100$ and n.s. $p > 0.9999$). **g** Quantifications of cell circularity for different genotypes (Kruskal–Wallis test from left to right: **** $p < 0.0001$, **** $p < 0.0001$ and * $p = 0.0129$). For (d-g), $n$(nub-Gal4) = 92 cells, $n$(*ft-PCP*) = 90, and $n$(*nub-Gal4>Patronin^RNAi*) = 90 cells and 3 pupae per genotype. **h** Representative images of 18 hAPF Patronin-overexpression (*nub-Gal4>Patronin-GFP*) stained for Arm to visualize cell shape. **i** Quantification of the cell major axis (along the P/D axis) upon Patronin-overexpression (two-tailed Mann-Whitney test: ** $p = 0.0032$). **j** Quantification of the cell minor axis (along the P/D axis) upon Patronin-overexpression (two-tailed t-test: *** $p = 0.0001$). **k** Quantification of the cell area upon Patronin-overexpression (two-tailed Mann-Whitney test: n.s. $p = 0.1685$). **l** Quantification of the cell circularity upon Patronin-overexpression (two-tailed Mann-Whitney test: **** $p < 0.0001$). For (**i–l**), $n$(nub-Gal4) = 92 and $n$(*nub-Gal4>Patronin-GFP*) = 110 and 3 pupae per genotype. For (**c–g**, **i–l**), boxes in the plot extend from the 25th to the 75th percentiles, with a line at the median. Whiskers show min and max values. Source data are provided as a Source Data file. Scale bars, (**a**, **h**) 5 μm.

the *ft-PCP* mutant and Patronin-depleted wings, the hinge contracts normally (Supplementary Fig. 6c). Hence, the initial cell elongation and consequent tissue elongation should be rescued if it depends on anisotropic stress that emerges from the hinge constriction. Since the tissue is shorter in both cases of genetic perturbation of microtubule organization (Fig. 5d; Supplementary Fig. 6b), these results provide direct experimental evidence that initial cell elongation is independent of extrinsic forces. The proposed model is consistent with published data showing that extrinsic mechanical forces act during the late phase of wing reshaping starting after 18 hAPF when the extracellular matrix

protein Dumpy becomes patterned at the wing margin[26,27]. Conceptually, it complements recent studies on the role of microtubule mechanics for large-scale shape changes[12,15,48,49]. Importantly, while this research underscores the significance of microtubule mechanics in wing development, we do not dismiss the possibility that other aspects of microtubules, such as polarized trafficking and/or the localization of transmembrane proteins, may also play a crucial role. Similarly, it does not mean that microtubule-based cell shape changes are the sole source of wing development. We consider the identified mechanism to be complemented by additional cellular events (e.g., cell intercalation/

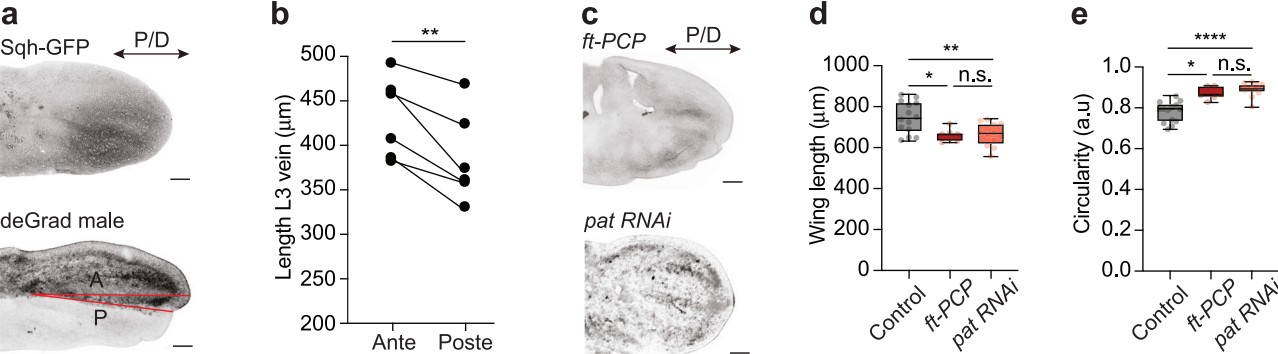

**Fig. 5 | Cell shape changes affect tissue shape. a** Representative images of control (*sqh^AX3^/Y; sqh-Sqh-GFP/tubGal80^ts^; ciGal4/+*) and deGradFP male (deGradFP (*sqh^AX3^/Y; sqh-Sqh-GFP/tubGal80^ts^; ciGal4/UAS-NSlmb-vhhGFP4*) wings at 18 hAPF. **b** Graph showing quantification of anterior and posterior wing lengths in the deGradFP male marked as indicated by red lines in **a** (two-tailed paired t-test, **$p = 0.0063$, $n = 6$ wings and 6 pupae). **c** Representative images of *ft-PCP* (*ft^l(2)ft^/ft^GRV^;act-Gal4/UAS-FtΔECDΔN-1*) mutant and Patronin-depleted (*nub-Gal4>Patronin^RNAi^*) wings at 18 hAPF. **d** Graph showing quantification of 18 hAPF control (*w^1118^*), *ft-PCP* mutant, and Patronin-depleted wing lengths (Ordinary one-way ANOVA: **$p = 0.076$, *$p = 0.0165$ and n.s. $p = 0.9842$, $N$ (wings) = 14, 7, 12). The tissue is shorter in the ft-PCP mutant and Patronin-depleted wings, where cells fail to elongate. **e** Graph showing quantification of 18 hAPF control (*w^1118^*), *ft-PCP* mutant, and Patronin-depleted wing circularity (Kruskal-Wallis test: ****$p < 0.0001$, *$p = 0.0132$ and n.s. $p > 0.9999$, $N$ (wings) = 14, 7, 12). For (**d, e**), boxes in the plot extend from the 25th to the 75th percentiles, with a line at the median. Whiskers show min and max values. Source data are provided as a Source Data file. Scale bars, (**a, c**) 50 μm.

T1 transition, convergent extension, polarized division, localized cell death, and changes in cytoplasm viscosity) that jointly shape wing tissue during morphogenesis.

Finally, our findings also have implications for planar cell polarity. As an encoder of spatial information, the Fat-PCP signaling pathway, which coordinates the planar polarization of cells within the tissue plane during various morphogenetic processes in invertebrates and vertebrates, is highly conserved. Fat-PCP consists of the atypical cadherins, Fat (Ft) and Dachsous (Ds), and the Golgi resident protein Fourjointed (Fj), a transmembrane kinase[50–53]. Ft and Ds interact across adherens junctions, forming heterodimers across adjacent cells that Fj modulates. As Fj and Ds expression both display gradients along the P/D axis, Ft–Ds heterodimers accumulate in a polarized fashion along the global axis. The net result is the conversion of tissue-wide transcriptional gradients of Fj and Ds into functional polarization of Ft-Ds at the cellular level, thereby providing a spatial cue to orient diverse developmental processes. Consistently, Ft-PCP was shown to orient cell divisions, tissue growth, cell rearrangements, and cell migration in *Drosophila*, zebrafish, and mammals[21,54–61]. Considering that aberrant PCP signaling yields a failure of tissue elongation, which leads to many developmental anomalies such as body truncation and neural tube defects, we propose that the microtubule cytoskeleton plays an important role in shaping cells during development and homeostasis. To determine if the observed mechanism may present a general feature of animal cell biology, it will be critical to elucidate the mechanical interplay of microtubules and actomyosin and its dependency on PCP signaling in other biological systems.

## Methods
### Drosophila melanogaster
The following mutant and transgenic fly strains were used in this study: *w^1118^* (BDSC 3605), *arm-Arm-GFP* (BDSC 3605), *sqh^AX3^; sqh-Sqh-GFP* (BDSC 57144), *ciGal4, actGal4, tubGal80^ts^* (BDSC 7019), *nubGal4* (BDSC 86108), *d^1^*(BDSC 270), *d^GC13^* (BDSC 6389), *dp^ov1^* (BDSC 276), *ft^GRV^* (BDSC 1894), *ft^l(2)fd^* (BDSC 1894), *Patronin* RNAi (BDSC 36659, HMS01547), *Dlg-GFP* (BDSC 59417), *UAS-alphaTub84B-EOS* (BDSC 51313) stocks from BDSC, *MyosinVI* RNAi stock from VDRC (VDRC 37534), *UAS-NSlmb-vhhGFP4 (UAS-deGradFP* construct) from D. Brunner, *UAS-GFP-DN-Zip* from D. Kiehart, *UAS-FtΔECDΔN1* from S. Blair, *UAS-GFP-Patronin* from M. Gonzalez-Gaitan. All fly stocks were raised at 25 °C (unless otherwise

mentioned) and grown on standard cornmeal-agar medium. The stocks are listed in Flybase (www.flybase.org).

### deGradFP expression and standardization of the dual temperature regime
Continuous expression of deGradFP at 25 °C was lethal for the embryos and did not allow growth until the desired stages of development. Therefore, to disrupt Myosin II contractility during the desired stages of pupal wing development, deGradFP must be expressed in a temporally controlled way using a *ciGal4* driver and tubGal80^ts^ system at a combination of 18 °C and 29 °C. The different time regimes of dual temperature were established to obtain the desired developmental stages where the role of Myosin II contractility was analyzed. The different dual temperature regimes, their equivalent developmental stages, and the assays that were performed using those regimes are summarized in Supplementary Table 1 (Supplementary information).

### Adult wing preparation
Wings from the adult flies were dissected and mounted on a glass slide in Canada balsam (Sigma-Aldrich) mixed with a tiny drop of absolute ethanol.

### Immunohistochemistry
White prepupae were collected at 0 hAPF from desired stocks and crosses growing either at 18 °C (for all deGradFP sets of experiments) or 25 °C (for all sets of experiments other than deGradFP). The prepupae collected at 0 hAPF were then either shifted to 29 °C (for all tubGal80^ts^ set of experiments) or grown at 25 °C (for all sets of experiments other than tubGal80^ts^) until the final desired developmental stages were reached. The pupae were fixed for 6–8 h in 4% PFA (with 0.1% Triton X-100) at room temperature. Wings were dissected and washed thoroughly with PBT (PBS + 0.1% Triton X-100) 2–3 times. Immunostaining was performed using standard protocols[32] with minor modifications. Primary antibody (or antibody cocktail with appropriate antibody dilution) was first diluted in PBTB (PBS + 0.1% Triton X-100 + 2% BSA) and then added to the wings. The wings were then incubated overnight (~16 h) at 4 °C on a rotor. After three washes using PBT secondary antibody diluted in PBTB was added to the wings. Wings were incubated for 2 h at room temperature and washed three

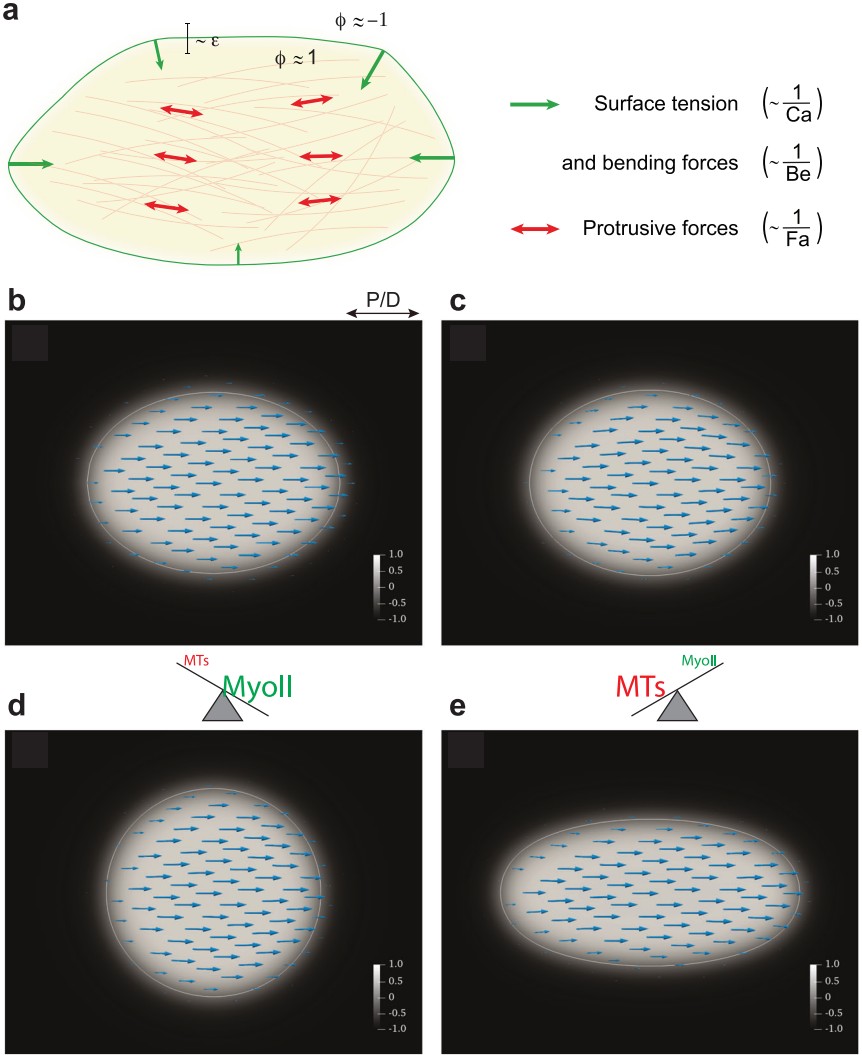

**Fig. 6 | Computational verification of the force-balance hypothesis. a** Schematic description of the computational model. The cell is modeled by the phase field variable φ that takes on the value 1 inside the cell (beige area) and −1 outside with a smooth transition in the interface region whose thickness is proportional to the parameter ε. The cell membrane is implicitly defined as the region where φ = 0 (green line). The microtubules (bleached red lines in the background) are not tracked individually but only through the orientation field $P$, which gives the average microtubule direction at each point. The protrusive force of the micro-tubules is modeled by active stress in the direction of the orientation field (red arrows). The contractile myosin forces are modeled by the surface tension and bending forces (green arrows), which minimize the surface curvature and area. **b** Initially (at time t = 0), the cell is chosen to be elliptic with microtubules oriented in the P/D (x) direction. **c** Cell shape at steady state (time t = 5) for counteracting protrusive and contractile (surface tension) forces. The computational parameters were chosen as Fa = −1 and Ca = 0.1. For this choice of parameters, the cell approximately maintains its shape, indicating a force balance. **d** If the protrusive force is reduced (Fa = −10, Ca = 0.1), the cell is significantly shorter in the P/D direction at a steady state. **e** If the surface tension is reduced instead (Fa = −1, Ca = 1), the cell is significantly elongated at a steady state. The color bar shows the value of the phase-field parameter φ, and turquoise arrows represent the orienta-tion field $P$ (average microtubule orientation). The white line indicates the cell membrane (zero level set of φ).

times in PBT. Finally, wings were mounted on glass slides in Vecta-shield with or without DAPI (Vector Laboratories) and covered with coverslips. For the third instar larval wing discs, 4% PFA (with 0.1% Triton X-100) was used to fix the samples for 40 min, and protocol for pupal wings was followed.

For immunofluorescence staining with pMRLC, pupae were fixed for 1 h in 18% PFA (with 0.1% Triton X-100) at room temperature. Wings were dissected and washed in PBT (PBS, 0.3% Triton X-100) three times (10 min each) followed by blocking for 1 h with PBTB (PBS, 0.3% Triton X-100, 0.5% BSA) and incubated overnight in primary antibody cocktail (in PBTB) at 4 °C. The wings were washed 4 times (10 min each) using PBT followed by incubation in a fluorescently conjugated secondary antibody cocktail (in PBTB, for 2 h, at room temperature) and again washed three times (20 min each) with PBT before mounting on glass slides in Vectashield with DAPI.

The following primary antibodies were used: rabbit anti-α-Tubulin (1:200; ab18251, Abcam); mouse anti-Armadillo (1:100; N2 7A1, DSHB); mouse anti-Flamingo (1:100; DSHB); rabbit anti-pMRLC (1:50; 3671 S, Cell Signaling Technology); rabbit anti-Dcp-1 (1:100; #9578, Cell Sig-naling Technology); rat anti-Tyr-Tubulin (1:750; ab6160, Abcam) and mouse anti- α-Tubulin (1:750; T9026, Sigma). Rhodamine phalloidin dye (1:100; Invitrogen) was used to visualize prehair orientation. Fluorophore-conjugated secondary antibodies (Invitrogen) were used at 1:200 dilution.

**Imaging of pupal wing**
To ensure reproducible results, we analyzed only a small area between wing veins L2 and L3, as depicted by the ROI in Supplementary Fig. 2a. We considered this necessary, as cells are smaller in the vein and inter-vein regions as well as in the proximal part of the wing pouch next to

the hinge compared to the rest of the wing blade. Also, cell elongation progresses as a wave from the posterior to the anterior side of the wing, yielding temporal differences in cell elongation across the wing. Wings were imaged using an upright LSM 710 confocal microscope (Carl Zeiss). The images were acquired using Zen software (Carl Zeiss, version 6.0, 2010) at different high magnification objectives for different experiments depending on the resolution needed. In general, 5x (0.16 EC Plan-Neofluar, Carl Zeiss) was used for all low magnification images to visualize whole wing morphology and determine the landmarks for precisely identifying developmental stages. For high magnification images, 40x (1.3 Oil C Plan-Apochromat, Carl Zeiss), 63x (1.4 Oil Plan-Apochromat, Carl Zeiss) and 100x (1.46 Oil α-Plan-Apochromat, Carl Zeiss) objectives were used. All high-magnification images were acquired at a step size of 0.3 μm (unless otherwise mentioned). For live imaging (Fig. 1 and Fig1S, control pupae expressed Arm-GFP or Ecad-GFP were collected at 14 hAPF, and the cuticle was gently removed to get the pupae out of their pupal cases. The pupae were placed laterally on a coverslip smeared with a thin layer of glue placed on top of a glass slide with spacers. Time-lapse 2D images with a frame rate of 100-300 ms were acquired using a 100x objective (1.46 NA Oil α-Plan-Apochromat, Carl Zeiss) mounted on an upright AxioImager.M2 microscope (Carl Zeiss) equipped with CSU10B spinning disk (Yokogawa) and a sCMOS ORCA Flash 4.0LT system (Hamamatsu). Images were acquired using VisiView software (Visitron Systems GmbH) after every 10 mins for 4 h.

### Imaging of adult wing
Adult wings were imaged using Imager.M1 microscope (Carl Zeiss) equipped with CoolSNAP ES2 camera (Photometrics) using 10x objective (0.3 EC Plan-Neofluar, Carl Zeiss).

### Transmission electron microscopy (TEM)
For TEM analysis, pupal wings of the appropriate age were fixed overnight at RT in a mixture of 2.5% glutaraldehyde in phosphate buffer (pH 7.3) and were further processed as described previously[11].

### Laser ablation
Pupae of desired genotypes and developmental stage corresponding to 18 hAPF were collected, and the cuticle was gently removed to get the pupae out of their pupal cases. The pupae were placed laterally on a coverslip smeared with a thin layer of glue placed on top of a glass slide with spacers. The pupae expressed Arm-GFP or Sqh-GFP to visualize cell junctions and actomyosin cables for ablation. A single-pulse of 355 nm laser (DPSL-355/14, Rapp OptoElectronics) at 2% laser power was used across a 10-pixel (0.64 μm) line perpendicular either to the center of cell junctions (aligned along P/D and A/P axes) or to the center of Myosin cable for ablation. Time-lapse 2D images with a frame rate of 100 ms were acquired using a 100x objective (1.46 NA Oil α-Plan-Apochromat, Carl Zeiss) mounted on an upright AxioImager.M2 microscope (Carl Zeiss) equipped with CSU10B spinning disk (Yokogawa) and an sCMOS ORCA Flash 4.0LT system (Hamamatsu). Images were acquired using VisiView software (Visitron Systems GmbH) from at least 1 minute before ablation and up to 4 to 5 minutes post-ablation to visualize the movement of cell junctions, vertices, and Myosin II cables before and after nanoablation.

For laser-induced hinge ablation, a single-pulse 355 nm laser (DPSL-355/14, Rapp OptoElectronics) at 2% laser power was used to ablate the hinge of one of the wings (roughly near the hinge-blade border) using a line ROI along its entire width such that the cells in the blade were mechanically uncoupled to the hinge. Time-lapse images at a time interval of 5 min between the frames were acquired using a 40x (1.3 NA Oil C Plan-Apochromat, Carl Zeiss) objective. Images were acquired at least 5–10 min before ablation until the end of hexagonal packing corresponding to 26 hAPF to visualize cell elongation (at an early stage) within minutes of hinge ablation and hexagonal packing of the cells (at a late stage) upon hinge ablation.

### Image processing
Images were processed and analyzed using Fiji/ImageJ software (NIH, version 2.0.0, 2015). Images of pupal and adult wings acquired in parts were stitched into whole wings using the Fiji plugin "pairwise stitching." For all the analyses related to apical cell shape, microtubule orientation, pMRLC signal intensity, and apical area, only the apical slices at the level of adherens junctions were z-projected. The z-slices at the level of adherens junctions in different images were determined by signals from Arm-GFP, anti-Armadillo antibody, anti-α-Tubulin antibody, anti-pMRLC antibody and Sqh-GFP. All the measurements were performed in Fiji using the "Analyze" feature. Brightness and contrast were adjusted within the linear range wherever needed. If used, the median filter of 0.5 or 1.0 was applied to all the images in a given set of analyses. The whole wing images were cropped in Fiji to represent appropriate ROIs wherever needed. Appropriate and desired fluorophore channels were merged or split from multi-channel images for representation as needed. Images were also converted into greyscale wherever needed for representation.

### Quantitative image analysis
For fixed cells, images were manually processed in Fiji for various parameters. All raw measurements were then summarized and further computed. Automated image analysis of live cell shape evolution during development (Fig. 1 and Supplementary Fig. 1) was done by custom scripts written in Jupyter Notebook. The source code has been deposited in GitHub [https://github.com/Die-Nase/Singh_2024_NatComm]. The total numbers of 'N' and 'n' analyzed for the measurements in all experiments are mentioned in the associated figure legends.

### Quantification of cell length
ROIs were manually drawn by tracing the apical cell outlines marked by Ecad-GFP, Arm-GFP, or anti-Armadillo antibody signals. "Feret's diameter," a Fiji function, was used to measure the length of cells and circularity.

### Quantification of wing length
ROIs were drawn manually by tracing the wing blade at each stage in wings marked by Ecad-GFP. "Feret's diameter", a Fiji function, was used to measure the length of the wing blade along the PD-axis.

### Quantification of cell height
The wings were imaged at each stage by acquiring Z stacks marked by Dlg-GFP. ROI tool was used to manually measure the apico-basal cross-section at different regions in each wing.

### Quantification of cell area
ROIs were drawn manually by tracing the apical cell outlines marked by Arm-GFP or anti-Armadillo antibody signals. The area was measured in square microns ($\mu m^2$) using the Fiji function "Analyze » Area."

### Quantification of pupal vein area
The images of the whole pupal wing were z-projected at the level of the maximum cross-sectional vein area. ROIs were drawn manually by tracing the edges around vein L2. The area was measured in square microns ($\mu m^2$) using the Fiji function "Analyze » Area".

### Quantification of microtubule intensity
Microtubule intensity was quantified by measuring mean intensity using the Fiji rectangle ROI function (7-8 cells per ROI) in wings stained with anti- α-tubulin antibody.

## Quantification of displacement and recoil velocity of junctions upon laser ablation

The movement of vertices associated with the ablated junctions was tracked post-ablation manually for 2 seconds at an interval of 200 milliseconds. The displacement was calculated by the distance between the two vertices every 200 milliseconds over a period of 2 seconds. This was represented as a displacement-time graph with mean and error values for a total of 10 measurements every 200 milliseconds. The recoil velocity was calculated for the initial 200 milliseconds (in the linear range) by dividing the initial displacement over the first 200 milliseconds. This initial recoil velocity was used as a proxy measurement for junction tension.

## Quantification of the angular distribution of microtubules

Appropriate apical slices were z projected. All the wings were aligned along their P/D axis. Angular or polarized distribution of microtubules was measured by the Fiji plugin "Orientation J." Microtubule orientation from three different wings of the same genotypes (100-120 cells per genotype) was pooled together. The distribution of microtubule orientation was plotted using polar plots in MATLAB (R2022b).

## Quantification of pMRLC polarity and intensity along the junctions

Appropriate apical slices were z projected. All the wings were aligned along their P/D axis. Polarization of pMRLC was measured by the Fiji plugin "Orientation J". pMRLC orientations were pooled together and binned into three categories of angular distribution (0–30°, 30–60° and 60–90°) with the P/D axis. The mean population of binned categories was compared. The quantification of the junctional pMRLC intensity was performed by measuring the mean intensity of a 3-pixel-thick line (corresponding to a 400 nm-wide stripe) using the Fiji linear ROI function along the junctions that were visualized by Arm staining. The background signal was subtracted from each of the intensity signals. Then the intensity values along each junction were normalized with respect to the average intensity signal of the control junctions in the same image (junctions on the posterior side of the wing).

## Quantification of cell shape (circularity)

ROIs were drawn manually by tracing the apical cell outlines using Arm-GFP or anti-Armadillo antibody signals. Cell circularity was measured using the Fiji function "Shape descriptors." Cell circularity is a measure of how close a cell is to being a perfect circle (circularity = 1). The mean cell circularity for many cells was computed and compared across different genotypes.

The formula for circularity is as follows:

$$Circularity = \frac{4\pi Area}{(perimeter)^2} \qquad (1)$$

## A computational model of actomyosin-microtubule force balance

The model was originally used to model cell motility by actomyosin contractile stress[62]. Here, we adopt this approach to model active forces generated by the microtubules (Fig. 3A). The cell is modeled as an active polar gel surrounded by a membrane that separates it from the surrounding extracellular fluid. The model uses a diffuse interface description of the cell, i.e., the cell is modeled by the phase-field parameter φ that takes on the value 1 inside the cell and −1 outside with a smooth transition in the interface region. The cell membrane is implicitly defined as the region where φ = 0. The average orientation of the microtubules is tracked by the vector-valued orientation field P. Note that P only represents the average direction of microtubules, not the microtubule density or the strength of the local pushing force. Hence, the model (weakly) enforces unit length |P| = 1 for the

orientation field vectors inside the cell and ensures that the orientation field vanishes (|P| = 0) outside the cell, with a smooth transition in the interface region. For the fluid, we track the velocity u and the pressure p using the Stokes equations. For simplicity, we assume equal density for the cytoplasm and the extracellular fluid.

The model equations are obtained by assuming that the system evolves according to a gradient descent of the free energy

$$E(\phi, P, u) = E_{kin}(u) + E_S(\phi) + E_P(\phi, P) \qquad (2)$$

which is composed of the kinetic energy of the fluid, the membrane energy $E_S$ and the energy of the microtubule network $E_P$. The membrane energy consists of a Helfrich-type bending energy[63] and the surface energy

$$\frac{1}{Ca} \int_X \frac{\epsilon}{2} |\nabla\phi|^2 + \frac{1}{\epsilon} W(\phi) dx \qquad (3)$$

corresponding to a classic Cahn-Hilliard model[64,65]. Here, $\epsilon$ is the phase-field parameter describing the thickness of the interface region and $W$ is a double-well potential with minima at −1 and 1. The capillary number $Ca$ regulates the strength of the surface tension (higher values of $Ca$ model lower surface tension). The bending resistance is controlled by the parameter $Be$.

The cell (phase field) and the microtubules are advected with the fluid flow and appear as additional stress terms in the Stokes equation. In particular, the protrusive force of the microtubules enters via the active stress term

$$\frac{1}{Fa} \widetilde{\phi} P \otimes P \qquad (4)$$

where $\widetilde{\phi} = \frac{1}{2}(\phi + 1)$ is an indicator function for the cell interior and the (negative) active force number $Fa$ controls the strength of the protrusive force (higher absolute value of $Fa$ means lower protrusive force). In our simulations, we keep $Be$ fixed and only vary $Ca$ and $Fa$, i.e., we assume a fixed bending resistance and model the actomyosin forces through the surface tension.

The resulting coupled nonlinear system of partial differential equations (PDEs) is solved using our discretization module dune-gdt (https://zivgitlab.uni-muenster.de/ag-ohlberger/dune-community/dune-xt/-/blob/master/README.md) based on the software framework DUNE[66]. For a detailed description of the model and the numerical approach, see[42].

## Statistical analyses

All the datasets presented in this work were first tested for normal distribution using D'Agostino & Pearson or Shapiro-Wilk normality tests. Statistical significance was determined using two-tailed unpaired/Student's t test or ordinary one-way ANOVA (with Tukey's multiple comparisons) when the data were distributed normally for two or more than two groups, respectively. Two-tailed Mann-Whitney U test or Kruskal-Wallis test (with Dunn's multiple comparisons) were performed when datasets were not distributed normally. All the graphs with scatter dot plots show mean values with red lines. Boxes in all box plots extend from the 25th to 75th percentiles, with a line at the median. Whiskers show min and max values. All the bar graphs indicate mean ± sd. The significance levels in all the graphs are as follows; n.s. (nonsignificant), * ($p \le 0.05$), ** ($p \le 0.01$), *** ($p \le 0.001$) and **** ($p \le 0.0001$). The statistical tests were performed using Prism7 (version 7.0d for Mac OS X, GraphPad Software). All experiments presented in the manuscript were repeated at least in three independent experiments/biological replicates. The experiments were not randomized, and the sample size was not predetermined.

## Reporting summary

Further information on research design is available in the Nature Portfolio Reporting Summary linked to this article.

## Data availability

All data are available in the main text or the supplementary materials. Source data are provided with this paper.

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

## Acknowledgements

We thank Damian Brunner, Daniel P. Kiehart, Seth S. Blair, Marcos Gonzalez-Gaitan, the Bloomington Stock Center and the Developmental Studies Hybridoma Bank for providing fly stocks and antibodies and Matis lab members for critically reading the manuscript. A.S., S.T., and M.M. were supported by the Cells-in-Motion Cluster of Excellence EXC-1003 (FF-2015-07) and the Deutsche Forschungsgemeinschaft (DFG, SPP-1782, MA 6726/3-1). T.L. and M.O. were supported by the DFG under Germany's Excellence Strategy EXC 1003 FF-2015-07 and EXC 2044 – 390685587, Mathematics Münster: Dynamics–Geometry–Structure. L.L. and M.G. were funded by the DFG (GA2268/3-1, GA2268/4-1). A.R., H.N. and J.K. acknowledge support from the DFG CRC1348/A02.

## Author contributions

A.S. and S.T. designed and performed the experiments, analyzed the data and wrote the manuscript. A.R., H.N. and J.K. prepared and imaged TEM samples. L.L. and M.G. wrote the script for the automated analysis of cell shape changes. T.L. and M.O. developed the model and carried out the simulation. M.M. supervised the research and wrote the manuscript with feedback from all authors.

## Funding

## Competing interests

The authors declare no competing interests.
