## [Peer Review File · Nature Communications]

Dynamic Interplay of Microtubule and Actomyosin Forces Drive Tissue ExtensionREVIEWER COMMENTS

Reviewer #1 (Remarks to the Author):

In this manuscript, Singh and colleagues build up on previous work where they characterised the cell autonomous role of microtubules and their orientation for regulating the elongation of cells (Singh et al. Nat Cell Biology 2018) in the Drosophila pupal wing. Here, they complement this work by exploring the role of microtubule reorganisation at earlier stage (from an apico-basal array to a medio-apical array preferentially oriented along proximo-distal axis) which correlates with dramatic cell shape changes (shortening along apico-basal axis and elongation along PD axis) and global wing elongation. They first exclude the contribution of external forces to this early elongation, and then use acute local perturbation of MyoII to exclude a function of MyoII in this cell elongation (which rather tends to reduce cell elongation through polarised tension along PD axis). Finally, they demonstrate that the polarised organisation of MTs has a direct impact on cell and tissue shape, since modulation of PCP or depletion of patronin (a – end binding MT that organises non centrosomal MTs) abolish the polarised distribution of MTs and the remodeling of the bundles along apico-basal axis, prevent cell elongation and directly impact wing elongation both at the pupal and adult stage. Eventually, they complement these observations by a relatively simple single cell model showing how polarised pushing forces generated by MTs and balanced by actomyosin (like the classic tensegrity model) can account for cell elongation.

These data are overall very interesting as this is one of the few reports documenting a clear remodeling of MTs organisation that directly impact cell shape which then modulate significantly organ shape, independently of actomyosin (or at least with a much minor role). Previous work has documented the impact of MT remodeling on morphogenesis, but they mostly focused on the impact of MTs on actomyosin regulation. There are only a limited number of works in vivo connecting MT reorganisation with cell shape changes that can be clearly disconnected from actomyosin remodeling (see for instance Takeda et al., NCB 2018, previous work from the same author, Singh et al., NCB 2018, or MTs role during cell extrusion Villars et al. Nat Comm 2022). As such, these results are important and will help to better establish the role of MTs as an important driver of cell shape and tissue morphogenesis independently of actomyosin, a point that could be quite important for the morphogenesis and epithelial large community.

The demonstration is overall convincing and the manuscript well written. There are however some results that remain a bit hard to interpret at this stage where additional controls would be needed, as well as additional measurements (which authors may already have in their hands). Finally, I would also suggest to be maybe more cautious on some statement by discussing more openly alternative interpretation in the discussion. With these additions (which only requires one more experiment and some quantifications as well as text editing) I would be very supportive for publication.

Main suggestions :

1. There are many strong statements in the manuscript, specially regarding the mechanical role of MTs

which is assumed to be based on pushing forces directly generated by MTs (thus in the line of the classic tensegrity models described by Dan Ingber some years ago). While all the data fit with this type of model, it is hard to exclude at this stage alternative indirect effects of MTs, for instance by polarised trafficking (which could affect membrane flux or the localisation of key transmembrane proteins), modulation of cytoplasm viscosity or modulation of nucleus shape. I don't think it is absolutely necessary to sort them, but mentioning that alternative model could explain the impact of MTs on cell shape in the discussion would be fair.

2. While the results are very interesting, the MyoII perturbation experiments are a bit hard to interpret at this stage. Indeed, while the cell gets slightly longer along PD axis, they also get much longer along AP axis, which cannot be explained with the single cell centered model of the authors. The best explanation is probably that the posterior compartment, where MyoII is still present, becomes relatively more tensed, and as such pull on the anterior compartment where MyoII is depleted hence elongation cells along AP axis (in line with this, cells look quite smaller in the posterior compartment upon MyoII depletion in the anterior domain, see Figure S2C, suggesting that there is indeed a tug of war). The results would be much easier to interpret if the authors could deplete MyoII homogeneously through the wing using for instance *nubbin gal4* (that covers both the hinge and the pouch). I expect that in this situation cells PD axis will lengthen, but cell elongation factor becomes also higher (long over short axis, see also my point below)

3. Cell geometrical measurements are essential for the demonstration in this article. However, many data should be included to have a full vision of cell shape changes, including total cell apical area, length along PD axis, length along AP axis, ratio of PD over AP axis length (cell elongation factor). If the authors have already all the cell contour, they can easily extract all these data in all their conditions. I would suggest to systematically include all of them, could it be for the description of the normal cell shape evolution, or during key perturbations. This will really help to fully describe cell shape changes. Of note, the authors use quite often the term "elongation" which I find quite confusing. Most commonly, elongation is not referring to an absolute length scale but the ratio of long over short axis of an object (which one can also call ellipticity), while this is quite often used in the manuscript to describe the evolution of the absolute cell length along PD axis. I would suggest to change the terminology to avoid confusion (for instance, talking about PD axis length versus elongation factor).

4. One important aspect of the argumentation is to show that MTS organisation precedes cell shape changes. This is indeed an important point since in many other systems cell shape dictates MT orientation and not the reverse (MT aligning along the cell long axis). The authors use in several instances the absolute PD length to make such argument, however I am not sure this measurement is the most relevant (since it could also change because of total area modulation). Instead, the best parameter would be the PD over AP axis length of the cell (the relative elongation along PD axis, or elongation factor). More specifically, in figure 1, the authors should show the evolution of this ratio over time (PD over AP cell length) which will help to compare with the main orientation of MTs. Cells seem indeed to have no preferential axis of elongation at 14h while MTs are already polarised which goes along the argument of the authors. This should be measured to prove that MTs orientation can be decoupled from cell long axis. Similarly, in Figure 3F, MTs have similar polarised orientation while cells are much less anisotropic/elongated upon MyoII depletion. Showing the PD over AP length in all these conditions and

compare it with MTs polarity will provide another argument for cell shape being downstream MTS and not the reverse.

5. The authors document quite extensively the variation of cell shape in various context which they correlate with wing shape. While this is probably valid, there might be over cellular events that contribute to the final wing elongation. Tissue lengthening can be decomposed in the contribution of cell shape changes, cell intercalation/T1 transition and convergent extension, polarised division and/or localised cell death (see for instance Etournay et al., *elife* 2015 <https://elifesciences.org/articles/07090>). To prove that cell shape changes are the main driver of the wing shape changes, one would need to completely exclude the contribution of these other factors, which could also be affected by PCP and MTs perturbations. In line with this, there is not perfect agreement between the elongation of cells along PD axis and the total elongation of the wing (see for instance Figure 1b, wing elongation of maybe 20%, and Figure 1g, >2 fold cell elongation). This may suggest either that cell lengthening is not homogeneous throughout the wing and/or that other cellular events buffer the impact of cell lengthening. I don't think the authors really need to document quantitatively all these factors (this remains a very time consuming task based on whole tissue segmentation), but at least they should acknowledge this limitation in the discussion and interpretation of their results.

Other minor points :

1. There are very relevant articles that may deserved to be included in the discussion. For instance, a similar MT reorganisation, but in the reverse order (from medioapical to apico-basal orientation) was described during salivary gland invagination in fly embryos and associated with cell apical constriction and elongation along apico-basal axis (see Booth et al *Dev Cell* 2014 <https://pubmed.ncbi.nlm.nih.gov/24914560/> and Gillars et al, *Nat Comm* 2021 <https://www.nature.com/articles/s41467-021-24332-0>). However, cell shape change is here driven by the impact of MTs on actomyosin organisation. Alternatively, MTs depletion was recently shown to promote cell apical constriction during cell extrusion independently of actomyosin modulation (Villars et al *Nat Comm* 2022 <https://pubmed.ncbi.nlm.nih.gov/35752632/>). This will help to position the authors article among other published work on cell shape modulation associated with MTs reorganisation.
2. Figure 1: the authors document a shrinkage of cells along apico-basal axis. Is it fully compensated by the increase of cell apical area ? (in other words, is the volume more or less preserved ?).
3. Figure 2d and g, lateral view : could the authors provide the z-scale bar on the figure ? Also, it is bit hard to see where is the basal limit of the cell in 2g. I guess the authors should be able to see cell limit by pushing the contrast and use this to draw the boundary on the panel ?
4. Figure 2k: could the authors try to find some metrics that could help to compare the level of MT polarisation between stages ? (maybe similar to MyoII, by binning angles and comparing the ratio of MT between -30/30 versus the rest ?). Qualitatively, it seems that MTs polarisation is getting stronger over time, which could be compatible with cell elongation enhancing as well MT polarity (but admittedly the authors have enough observations to show that cell shape is not sufficient to dictacte MT orientation).

5. Figure 4 and S4: could the authors provide an estimate of cell length along apico-basal axis upon PCP perturbation and patronin depletion ? While this is clearly not the main point of this article, if the MT maintenance along apico-basal axis in these perturbed conditions is also associated with more columnar cells, that would give another argument for MTs being instructive for cell axis elongation.

6. I could not find information on the positioning of the ROIs used for cell parameter measurement in the wing. It might be good to provide details on this (since there might be spatial differences among regions of the wing). The methods state that the ROIs are always shown on the corresponding wing but unless I am mistaken I could not find them in the figures.

7. Figure 1h legend : circularity (along PD axis), I guess "along PD axis" can be removed (since this is not relevant for circularity).

Reviewer #2 (Remarks to the Author):

Summary: The manuscript presents an argument for the mechanical drivers of a particular part of morphogenesis in *Drosophila*, namely the epithelium elongation of the wing. During wing extension, the cells in the tissue undergo a columnar-to-cuboidal shape remodeling. The authors ask whether this morphological change is driven by cell-cell interactions, actomyosin contractility, or microtubule dynamics. Through knockdowns and mutants, the authors show that 1) E-cadherins do not appear to be required for elongation, 2) elongation is also not associated with changes in myosin levels (knockdown deGradFP results in rounder cells in Fig.3), 3) disrupting microtubule organization (or polarity) halts cell elongation and ultimately results in a smaller, rounder wing. A mathematical model supports the finding that increased myosin activity (inward global stress) results in rounder cells while outward tangential/lateral only stress associated with microtubules results in elongated cells. The observation, nicely extracted in order to observe the mechanical drivers of morphogenesis, is not surprising — increased isotropic pulling forces, due to myosin contractility, would make cells rounder while localized lateral protrusive forces, due to MT organization, would make cells more elongated. The connection between microtubules generating protrusive (rather than pulling forces) seems unclear — it is however very clear that MT organization is needed for elongation. Would have liked to see an investigation of the pushing rather than pulling forces exerted by MT.

Minor comments:

** It would be useful to see a 3D schematic of the cells in the developing *Drosophila* wing to explain the side and top-view.

** At times, the authors are not clear if by circularity they mean circularity of cells or the developing wing. This should be clarified in the text and in the figure captions.

** Why are other types of cadherins not visualized/considered (e.g. VE-cadherins). And why not perturb cadherins?

** Consider a subfigure in Fig.2 with a schematic needed to explain the angle of the MT orientation.

** Is the MT re-arrangement from 14 hAPF to 16 hAPF driven by extrinsic factors? It appears to me that this discussion only holds at 16-18 hAPF, correct?

** Cell proliferation does not occur on this timescale, I imagine?

** In other systems, supracellular actin cables have been implicated in driving elongation. Have the authors looked at actin arrangement?

** In the model must assume asymmetric/anisotropic distribution of protrusive forces (more in the lateral direction). Model seems too complicated to capture the difference between localized pushing vs global pulling forces (could have used a simple elastic cell membrane with contractile forces and lateral protrusive forces instead of a two-phase description)

** line 389 Ft-PCP instead of Fat-PCP

Reviewer #3 (Remarks to the Author):

The authors examine how cell intrinsic shape changes lead to global changes in tissue morphology, in this instance the pupal wing. In recent years there has been a focus on how extrinsic and intrinsic factors help shape a tissue using the pupal wing as a model system, and this work complements this research nicely. The authors identify that cell intrinsic shape changes during 14 to 18hAPF contribute significantly to the overall shape of the wing. Furthermore, the authors identify a role of microtubule reorganisation switching from an apical-basal alignment within the cell to a purely apical localisation within the cell and aligned along the proximal-distal (PD) axis of the developing wing. The authors identify the ft/ds planar cell polarity complex as critical for the alignment of microtubules along the PD-axis. This paper sheds interesting light on how intrinsic factors contribute to the shape of a tissue when linked to extrinsic cues, and the importance of microtubules in this process.

The experimental methodology used throughout the study is of a high-standard, and the microscopy of microtubules is of special note and a real highlight of the paper. I do not have the expertise to comment on the mathematical modelling directly, but the overall design is clear as a method to validate the model proposed. The current model from the authors is one where reorganisation of microtubules leads to a protrusive force that elongates the cells along the PD axis. However, from the experimental data presented it is not clear whether microtubules do indeed generate a protrusive force.

From the sqh experiments, lowering cortical tension leads to an increase in cell apical area, and therefore myosin activity acts as a negative pressure on apical area. The pat/ft experiments highlight that if the microtubules are not orientated then cells do not elongate along the PD-axis. An alternative model that can explain these findings and not rely on microtubules producing a protrusive force is one of shape instability and non-compressive elements. For instance, cell shape could fluctuate due to cortical tension across all cells being in a non-equilibrated state. Microtubules could therefore act as non-compressive struts that limit a reduction in apical area as cells contract in response to being deformed by neighbouring cells. As microtubules are polarised in the WT this would bias elongation along the PD-axis whilst in the ft and pat animals no bias is generated.

Based on that rationale for publication, I would ask the authors to either:

- Tone down their conclusions that microtubules generate protrusive forces to a more nuanced conclusion that aligned microtubule dynamics are important for cell shape changes, but their exact role is unclear.
- Or provide further experimental evidence that microtubules do generate protrusive forces.

For the latter, I would suggest as possible experiments:

- Live-image cell junctions at a fast (order of seconds) temporal resolution to determine how stable cell shape is. If the position of junctions does not fluctuate then one can assume that there is no dynamic instability within the system.
- Through laser ablation, excise out clusters of cells and examine their shape change. Initially one would expect an immediate reduction in apical area as tension across the system is released. However, after this immediate response, cells will then recover to a new state, and if microtubules generate protrusive forces the cells should elongate along the axis where microtubules are aligned. This type of experiment has already been done by the authors to some extent when they laser-dissected the hinge from blade at 18hAPF and observed no further cell elongation. However, smaller cuts as designed in Bonnet et al 2012 (<https://royalsocietypublishing.org/doi/epdf/10.1098/rsif.2012.0263>) would be far more instructive.
- Modulate the stability/force generation of microtubules without altering their alignment. Whilst the authors attempted this with the pat RNAi unfortunately the alignment of microtubules was also disrupted. If the authors could increase microtubule stability and show an increase in cell elongation as their model suggests, this would provide direct evidence to support their hypothesis.

Minor issues:

I have an issue with the measure of circularity, as a proxy for cell elongation. As circularity is a measure of how close to a perfect circle an object's shape is, a low value doesn't necessarily suggest that cells are elongated. Whilst I doubt this would change the results drastically, as it is doubtful any star shape cells exist, I would advise the authors based on the other shape descriptors they quantified (major and minor axis lengths) to use $1 - \text{minor axis}/\text{major axis}$ as a better read-out for cell elongation.

Could the authors include a measurement of apical area to all perturbations and not only major and minor axis lengths, this would remove any ambiguity of whether apical area changes could explain differences in total wing length.

Could the authors either please include PD-axis guides for each image panel or highlight in the text that all images are aligned with PD-axis along the horizontal axis.

Could the authors please include an analysis in the WT examining the orientation of each cell's major axis with respect to the PD axis, as highly variable orientation would put doubt into the contribution elongation makes to the overall tissue size.

Point-to-point rebuttal for manuscript NCOMMS-23-39767-T

We would like to start by thanking the reviewers for their detailed comments and constructive critique. The referees have identified important areas that indeed required improvement. After completing the suggested experiments, we are now happy to submit our revised manuscript.

Below, you will find a detailed point-by-point rebuttal with original comments in black and our response in blue.

Reviewer #1 (Remarks to the Author):

In this manuscript, Singh and colleagues build up on previous work where they characterised the cell autonomous role of microtubules and their orientation for regulating the elongation of cells (Singh et al. Nat Cell Biology 2018) in the Drosophila pupal wing. Here, they complement this work by exploring the role of microtubule reorganisation at earlier stage (from an apico-basal array to a medio-apical array preferentially oriented along proximo-distal axis) which correlates with dramatic cell shape changes (shortening along apico-basal axis and elongation along PD axis) and global wing elongation. They first exclude the contribution of external forces to this early elongation, and then use acute local perturbation of MyoII to exclude a function of MyoII in this cell elongation (which rather tends to reduce cell elongation through polarised tension along PD axis). Finally, they demonstrate that the polarised organisation of MTs has a direct impact on cell and tissue shape, since modulation of PCP or depletion of patronin (a – end binding MT that organises non centrosomal MTs) abolish the polarised distribution of MTs and the remodeling of the bundles along apico-basal axis, prevent cell elongation and directly impact wing elongation both at the pupal and adult stage. Eventually, they complement these observations by a relatively simple single cell model showing how polarised pushing forces generated by MTs and balanced by actomyosin (like the classic tensegrity model) can account for cell elongation.

These data are overall very interesting as this is one of the few reports documenting a clear remodeling of MTs organisation that directly impact cell shape which then modulate significantly organ shape, independently of actomyosin (or at least with a much minor role). Previous work has documented the impact of MT remodeling on morphogenesis, but they mostly focused on the impact of MTs on actomyosin regulation. There are only a limited number of works in vivo connecting MT reorganisation with cell shape changes that can be clearly disconnected from actomyosin remodeling (see for instance Takeda et al., NCB 2018, previous work from the same author, Singh et al., NCB 2018, or MTs role during cell extrusion Villars et al. Nat Comm 2022). As such, these results are important and will help to better establish the role of MTs as an important driver of cell shape and tissue morphogenesis independently of actomyosin, a point that could be quite important for the morphogenesis and epithelial large community.

The demonstration is overall convincing and the manuscript well written. There are however some results that remain a bit hard to interpret at this stage where additional controls would be needed, as well as additional measurements (which authors may already have in their hands). Finally, I would also suggest to be maybe more cautious on some statement by discussing more openly alternative interpretation in the discussion. With these additions

(which only requires one more experiment and some quantifications as well as text editing) I would be very supportive for publication.

We thank the reviewer for the positive comments about our manuscript and for acknowledging the underestimated role of microtubules in morphogenesis.

Main suggestions:

1. There are many strong statements in the manuscript, specially regarding the mechanical role of MTs which is assumed to be based on pushing forces directly generated by MTs (thus in the line of the classic tensegrity models described by Dan Ingber some years ago). While all the data fit with this type of model, it is hard to exclude at this stage **alternative** indirect effect of MTs, for instance by polarised trafficking (which could affect membrane flux or the localisation of key transmembrane proteins), modulation of cytoplasm viscosity or modulation of nucleus shape. I don't think it is absolutely necessary to sort them, but mentioning that alternative model could explain the impact of MTs on cell shape in the discussion would be fair.

Answer: We agree with the reviewer that microtubules may also affect cell shape through additional mechanisms besides the mechanical role discussed in this study. Following the referee's recommendation, we made following changes:

- Considering the lack of direct evidence, we changed the title and added a discussion where clearly state limitations of the current study to identify protrusive forces that originate directly from MTs.
- We expanded the discussion to include a more thorough treatment of these issues, also mentioning alternative mechanisms that may contribute to cell and tissue elongation (see also Point 5, below).

2. While the results are very interesting, the MyoII perturbation experiments are a bit hard to interpret at this stage. Indeed, while the cell gets slightly longer along PD axis, they also get much longer along AP axis, which cannot be explained with the single cell centered model of the authors. The best explanation is probably that the posterior compartment, where MyoII is still present, becomes relatively more tensed, and as such pull on the anterior compartment where MyoII is depleted hence elongation cells along AP axis (in line with this, cells look quite smaller in the posterior compartment upon MyoII depletion in the anterior domain, see Figure S2C, suggesting taht there is indeed a tug of war). The results would be much easier to interpret if the authors could deplete MyoII homogeneously through the wing using for instance nubbin gal4 (that covers both the hinge and the pouch). I expect that in this situation cells PD axis will lengthen, but cell elongation factor becomes also higher (long over short axis, see also my point below).

Answer: We thank the reviewer for bringing up this valid point. As suggested, we used alternative myosin perturbation methods to probe the role of MyoII more thoroughly. Specifically, we tried to disrupt MyoII function using deGradFP (i.e. MyoII light chain depletion), RNAi (i.e. MyoII downregulation), and optogenetically (i.e. employing the CRY2/CIBN system to regulate PI(4,5)P2 levels and subsequently modulate actomyosin

contractility). We combine these with tools for spatial and temporal control over perturbation, including nubGal4, ciGal4, and TubGal80 drivers. Following, we summarize all experimental results that were obtained:

- Conclusive data: In our hands, only the combination of deGradFP with ciGal4 and TubGal80 proved effective in reducing MyoII efficiently and within a short time window. We consider a short interval essential, as it minimizes interference with other morphogenic processes, such as cell division.
- Inconclusive data: Using nubGal4 alone, pupal wings failed to develop. We reason that downregulation from the beginning of wing disc development is not well suited, as it likely impacts cell divisions. To address this issue, we combined nubGal4 with TubGal80 and subjected pupae to a temperature shift at late third instar. Under these conditions, we observed no depletion of MyoII (i.e. MyoII-GFP and PMLC). We refrained from an earlier shift to 29°C (i.e. third instar larvae stage) to avoid affecting cell division. Additionally, a prolonged period of MyoII depletion would likely influence cell mechanics, rendering the results inconclusive.

3. Cell geometrical measurements are essential for the demonstration in this article. However, many data should be included to have a full vision of cell shape changes, including total cell apical area, length along PD axis, length along AP axis, ratio of PD over AP axis length (cell elongation factor). If the authors have already all the cell contour, they can easily extract all these data in all their conditions. I would suggest to systematically include all of them, could it be for the description of the normal cell shape evolution, or during key perturbations. This will really help to fully describe cell shape changes. Of note, the authors use quite often the term “elongation” which I find quite confusing. Most commonly, elongation is not referring to an absolute lengthscale but the ratio of long over short axis of an object (which one can also call ellipticity), while this is quite often used in the manuscript to describe the evolution of the absolute cell length along PD axis. I would suggest to change the terminology to avoid confusion (for instance, talking about PD axis length versus elongation factor).

Answer: This is a very important point. In the revised manuscript, we included more cell descriptors, and also expanded the analysis to live cells to probe for shape changes in a more fine-grained manner as follows:

- We show that the cell area is not changing over time, thus demonstrating that the observed increase in cell length along the P/D axis is caused by elongation and not due to overall cell growth.
- We also added an analysis of the minor axis length, which was missing in the initial submission, to Fig 1. Here, we find a reduction in the minor axis concomitantly with the increase of the major axis. As suggested, we also changed the terminology.
- Finally, we added a detailed analysis of live cell shape changes in the developing wing between 14.5 to 18.0 h APF (Supplementary Figure 1).

Collectively, we believe that this expanded parameter set (more variables, more time points) now allows a better characterization of the dynamic shape changes cells undergo during wing development.

4. One important aspect of the argumentation is to show that MTS organisation precedes cell shape changes. This is indeed an important point since in many other systems cell shape dictate MT orientation and not the reverse (MT aligning along the cell long axis). The authors use in several instance the absolute PD length to make such argument, however I am not sure this measurement is the most relevant (since it could also change because of total area modulation). Instead, the best parameter would be the PD over AP axis length of the cell (the relative elongation along PD axis, or elongation factor). More specifically, in figure 1, the authors should show the evolution of this ratio over time (PD over AP cell length) which will help to compare with the main orientation of MTs. Cells seem indeed to have no preferential axis of elongation at 14h while MTs are already polarised which goes along the argument of the authors. This should be measured to prove that MTs orientation can be decoupled from cell long axis. Similarly, in Figure 3F, MTs have similar polarised orientation while cells are much less anisotropic/elongated upon MyoII depletion. Showing the PD over AP length in all this conditions and compare it with MTs polarity will provide another argument for cell shape being downstream MTS and not the reverse.

Answer: We agree with the reviewer that cell shape descriptions are a key to understanding the molecular mechanism leading to changes. As discussed in the previous point, we expanded our analysis now showing for all conditions (i) area, (ii) cell major (i.e. P/D) axis, (iii) cell minor (i.e. A/P) axis and (iv) circularity.

5. The authors document quite extensively the variation of cell shape in various context which they correlate with wing shape. While this is probably valid, there might be over cellular events that contribute to the final wing elongation. Tissue lengthening can be decomposed in the contribution of cell shape changes, cell intercalation/T1 transition and convergent extension, polarised division and/or localised cell death (see for instance Etournay et al., *elife* 2015 <https://elifesciences.org/articles/07090>). To prove that cell shape changes are the main driver of the wing shape changes, one would need to completely exclude the contribution of these other factors, which could also be affected by PCP and MTs perturbations. In line with this, there is not perfect agreement between the elongation of cells along PD axis and the total elongation of the wing (see for instance Figure 1b, wing elongation of maybe 20%, and Figure 1g, >2 fold cell elongation). This may suggest either that cell lengthening is not homogeneous throughout the wing and/or that other cellular events buffer the impact of cell lengthening. I don't think the authors really need to document quantitatively all these factors (this remains a very time-consuming task based on whole tissue segmentation), but at least they should acknowledge this limitation in the discussion and interpretation of their results.

Answer: We agree with the referee that, while highly desirable, a comprehensive understanding of spatiotemporal shape changes of all cells in the developing wing, including cell intercalation/T1 transition, convergent extension, polarized division, and localized cell death, is beyond the scope of this single manuscript. As mentioned above, we acknowledge these limitations in the revised discussion.

Concerning the discrepancy in cell elongation and wing length, we think that one of the reasons for the disagreement can be attributed to the heterogeneity of the wing tissue. Indeed, to ensure reproducible results, we analyzed only a small area between wing veins L2 and L3 (ROI is marked now in Supplementary Figure 2a). We considered this necessary, as cells are smaller in the vein and inter-vein regions as well as in the proximal part of the wing pouch next to the hinge compared to the rest of the wing blade. Also, cell elongation progresses as a

wave from the posterior to the anterior side of the wing, yielding temporal differences in cell elongation across the wing. A detailed description of where images were captured has been added to the Materials section.

Other minor points:

1. There are very relevant articles that may deserved to be included in the discussion. For instance, a similar MT reorganisation, but in the reverse order (from medioapical to apico-basal orientation) was described during salivary gland invagination in fly embryos and associated with cell apical constriction and elongation along apico-basal axis (see Booth et al Dev Cell 2014 <https://pubmed.ncbi.nlm.nih.gov/24914560/> and Gillars et al, Nat Comm 2021 <https://www.nature.com/articles/s41467-021-24332-0>). However, cell shape change is here driven by the impact of MTs on actomyosin organisation. Alternatively, MTs depletion was recently shown to promote cell apical constriction during cell extrusion independently of actomyosin modulation (Villars et al Nat Comm 2022 <https://pubmed.ncbi.nlm.nih.gov/35752632/>). This will help to position the authors article among other published work on cell shape modulation associated with MTs reorganisation.

Answer: We thank the referee for pointing out this omission. The articles have been added to the revised manuscript.

2. Figure 1: the authors document a shrinkage of cells along apico-basal axis. Is it fully compensated by the increase of cell apical area? (in other words, is the volume more or less preserved?).

Answer: Measurements of cell height in Figure 1d were performed at cell-cell boundaries. Considering the non-uniform distribution of endogenous basolateral marker Dlg-GFP, which precludes an accurate measurement of the 3D cell shape (e.g., potential rounding of apical and basal membranes along the z-axis), we cannot decisively answer this question. To account for this limitation, we replaced in the figure 'cell length' with 'Dlg-GFP (A/B length, μm).

3. Figure 2d and g, lateral view : could the authors provide the z-scale bar on the figure ? Also, it is bit hard to see where is the basal limit of the cell in 2g. I guess the authors should be able to see cell limit by pushing the contrast and use this to draw the boundary on the panel?

Answer: We thank the referee for pointing this out. The basal boundary was marked.

4. Figure 2k: could the authors try to find some metrics that could help to compare the level of MT polarisation between stages? (maybe similar to MyoII, by binning angles and comparing the ratio of MT between -30/30 versus the rest?). Qualitatively, it seems that MTs polarisation is getting stronger over time, which could be compatible with cell elongation enhancing as well MT polarity (but admittedly the authors have enough observations to show that cell shape is not sufficient to dictacte MT orientation).

Answer: We analyzed the data, as suggested by the referee, and included the p values in the legends.

5. Figure 4 and S4: could the authors provide an estimate of cell length along apico-basal axis upon PCP perturbation and patronin depletion? While this is clearly not the main point of this article, if the MT maintenance along apico-basal axis in these perturbed conditions is also associated with more columnar cells, that would give another argument for MTs being instructive for cell axis elongation.

Answer: Due to the limitation of genetic combination, we could not use the endogenous basolateral marker Dlg-GFP. Instead, immunostaining with the Dlg antibody was performed. Unfortunately, the signal quality was insufficient for marking the basal side. This made it difficult to estimate the cell length along the apico-basal axis upon MT perturbations.

6. I could not find information on the positioning of the ROIs used for cell parameter measurement in the wing. It might be good to provide details on this (since there might be spatial differences among regions of the wing). The methods state that the ROIs are always shown on the corresponding wing but unless I am mistaken, I could not find them in the figures.

Answer: We thank the referee for pointing out this. We now added the box showing the area that was analyzed in all conditions (Supplementary Figure 2a).

7. Figure 1h legend: circularity (along PD axis), I guess "along PD axis" can be removed (since this is not relevant for circularity).

Answer: True. This has been corrected.

Reviewer #2 (Remarks to the Author):

Summary: The manuscript presents an argument for the mechanical drivers of a particular part of morphogenesis in *Drosophila*, namely the epithelium elongation of the wing. During wing extension, the cells in the tissue undergo a columnar-to-cuboidal shape remodeling. The authors ask whether this morphological change is driven by cell-cell interactions, actomyosin contractility, or microtubule dynamics. Through knockdowns and mutants, the authors show that 1) E-cadherins do not appear to be required for elongation, 2) elongation is also not associated with changes in myosin levels (knockdown deGradFP results in rounder cells in Fig.3), 3) disrupting microtubule organization (or polarity) halts cell elongation and ultimately results in a smaller, rounder wing. A mathematical model supports the finding that increased myosin activity (inward global stress) results in rounder cells while outward tangential/lateral only stress associated with microtubules results in elongated cells. The observation, nicely extracted in order to observe the mechanical drivers of morphogenesis, is not surprising — increased isotropic pulling forces, due to myosin contractility, would make cells rounder while localized lateral protrusive forces, due to MT organization, would make cells more elongated. The connection between microtubules generating protrusive (rather than pulling forces) seems unclear — it is however very clear that MT organization is needed for elongation. Would have liked to see an investigation of the pushing rather than pulling forces exerted by MT.

We thank the reviewer for the very positive comments about our work and the constructive suggestions.

Minor comments:

** It would be useful to see a 3D schematic of the cells in the developing *Drosophila* wing to explain the side and top-view.

Answer: We thank the reviewer for these suggestions. To improve clarity, we revised and expanded explanatory 2D schematics of tissue and cells during wing development (see Figure 1 and Figure 2). As suggested by the referee, we also tried to come up with a 3D schematic. However, the final drawing was so convoluted that the key message was lost. As we (repeatedly) failed to come up with a solution that was at the same time clear and reductionist, we did not include it in the revised manuscript.

** At times, the authors are not clear if by circularity they mean circularity of cells or the developing wing. This should be clarified in the text and in the figure captions.

Answer: We apologize for this omission. A definition was added to the figure legends.

** Why are other types of cadherins not visualized/considered (e.g. VE-cadherins). And why not perturb cadherins?

Answer: The referee raises two important questions, which we would like to answer sequentially.

- Visualize others cadherins: *Drosophila* has 3 classical cadherins (one E- and two N-cadherins, from a total of 17 proteins in the genome with cadherin repeats). From our transcriptome analysis, we know that DE-cadherin (Shg) is the major expressed cadherin in the wing at the 18 h APF (see below, Fig1). For that reason, we settled on this one.

- Perturb cadherins: We agree with the reviewer that a thorough characterization of all cadherins would undoubtedly provide exciting new insights. Considering that the main findings of this manuscript are centered on the role of microtubules, we respectfully argue that a comprehensive analysis of cadherins is beyond the scope of the present paper.

** Consider a subfigure in Fig.2 with a schematic needed to explain the angle of the MT orientation.

Answer: We thank the referee for this suggestion. The schematic was added.

** Is the MT re-arrangement from 14 hAPF to 16 hAPF driven by extrinsic factors? It appears to me that this discussion only hold at 16-18 hAPF, correct?

Answer: Regarding the control of microtubule alignments within the plane, we show that it depends on the Ft-PCP signaling pathway. The reorganization from apical-basal to planarly polarized starts at around 14 h APF, and within half an hour, microtubules are aligned along the planar axis (at 15 h APF only along the planar axis). Consistently, in mutant wings, where Ft-PCP is perturbed, microtubules stay aligned perpendicular to the planar axis. Thus, the Ft-PCP signaling pathway plays a crucial role in governing microtubule organization starting from around 14 hours APF. An unanswered aspect relates to the mechanism orchestrating microtubule alignment along the apical-basal axis before that, during the period between 9 hours APF and 14 hours APF.

** Cell proliferation does not occur on this timescales, I imagine?

Answer: The last round of cell division in pupal wings starts at around 15/16 h APF, and happens initially at low frequency. The majority of cell divisions happen after 18 h APF. We fully agree that an analysis of different cellular events during wing elongation (cell shape

changes, cell intercalation/T1 transition and convergent extension, polarised division and/or localized cell death) would be very informative, but we are limited both experimentally and within the scope of this single manuscript (see also reviewer 1, point 5). In addition, as shown by Etournay et al., Elife 2015 <https://elifesciences.org/articles/07090>), the majority of tissue strain at 15-18h APF can be explained by cell shape changes (i.e., cell elongation) excluding important contribution of cell division and/or T1 transition.

** In other systems, supracellular actin cables have been implicated in driving elongation. Have the authors looked at actin arrangement?

Answer: We thank the reviewer for the suggestions. It is true that in many systems, cell shape changes are driven by so-called supracellular actomyosin cables. These are the structures that we observe in our cells and that start to get prominent at later stages, especially after 18 h APF (most probably as a response to hinge contraction). To determine the role of these cables, we perturbed MyoII function as MyoII is the motor protein that is necessary for the pilling of the cables on cell junction. We also looked at actin specifically in the wing cells (by using phalloidin staining), but it shows the same localization as MyoII.

** In the model must assume asymmetric/anisotropic distribution of protrusive forces (more in the lateral direction). Model seems too complicated to capture the difference between localized pushing vs global pulling forces (could have used a simple elastic cell membrane with contractile forces and lateral protrusive forces instead of a two-phase description)

Answer: We agree with the referees that alternative models could be used. The possibility of using other models has been added to the text. Considering the consistency of the numerical model with the experimental data, we deemed the model suitable to support the main findings of the manuscript.

** line 389 Ft-PCP instead of Fat-PCP

Answer: This has been corrected.

Reviewer #3 (Remarks to the Author):

The authors examine how cell intrinsic shape changes lead to global changes in tissue morphology, in this instance the pupal wing. In recent years there has been a focus on how extrinsic and intrinsic factors help shape a tissue using the pupal wing as a model system, and this work complements this research nicely. The authors identify that cell intrinsic shape changes during 14 to 18hAPF contribute significantly to the overall shape of the wing. Furthermore, the authors identify a role of microtubule reorganisation switching from an apical-basal alignment within the cell to a purely apical localisation within the cell and aligned along the proximal-distal (PD) axis of the developing wing. The authors identify the ft/ds planar cell polarity complex as critical for the alignment of microtubules along the PD-axis. This paper sheds interesting light on how intrinsic factors contribute to the shape of a tissue when linked to extrinsic cues, and the importance of microtubules in this process.

The experimental methodology used throughout the study is of a high-standard, and the microscopy of microtubules is of special note and a real highlight of the paper. I do not have the expertise to comment on the mathematical modelling directly, but the overall design is clear as a method to validate the model proposed. The current model from the authors is one where reorganisation of microtubules leads to a protrusive force that elongates the cells along the PD axis. However, from the experimental data presented it is not clear whether microtubules do indeed generate a protrusive force.

From the sqh experiments, lowering cortical tension leads to an increase in cell apical area, and therefore myosin activity acts as a negative pressure on apical area. The pat/ft experiments highlight that if the microtubules are not orientated then cells do not elongate along the PD-axis. An alternative model that can explain these findings and not rely on microtubules producing a protrusive force is one of shape instability and non-compressive elements. For instance, cell shape could fluctuate due to cortical tension across all cells being in a non-equilibrated state. Microtubules could therefore act as non-compressive struts that limit a reduction in apical area as cells contract in response to being deformed by neighbouring cells. As microtubules are polarised in the WT this would bias elongation along the PD-axis whilst in the ft and pat animals no bias is generated. Based on that rationale for publication, I would ask the authors to either:

- Tone down their conclusions that microtubules generate protrusive forces to a more nuanced conclusion that aligned microtubule dynamics are important for cell shape changes, but their exact role is unclear.
- Or provide further experimental evidence that microtubules do generate protrusive forces.

We thank the reviewer for the positive comments and thoughtful suggestions.

Concerning the two options: we decided to tone down the strength of our statements. To that end, we (i) changed the title and (ii) clearly state limitations of the current study to identify protrusive forces that originate directly from MTs, (iii) discuss the potential contribution of other parameters.

Addendum: While we decided to weaken the strength of our statements, we acknowledge the elegant experiments suggested by the referee. Our replies are summarized in the following section.

For the latter, I would suggest as possible experiments:

- Live-image cell junctions at a fast (order of seconds) temporal resolution to determine how stable cell shape is. If the position of junctions does not fluctuate then one can assume that there is no dynamic instability within the system.

Answer: Using a linescan, we probed for shape stability using a confocal spinning disc microscope. Using an acquisition speed of 50ms/frame, we do not see any fluctuations of the cell cortex (see Figure below). However, as the local fluctuations needed for the insertion of the tubulin α/β heterodimer are likely beyond the spatio-temporal resolution of the confocal microscope, we cannot decisively comment on this and will therefore not discuss it in the paper.

- Through laser ablation, excise out clusters of cells and examine their shape change. Initially, one would expect an immediate reduction in apical area as tension across the system is released. However, after this immediate response, cells will then recover to a new state, and if microtubules generate protrusive forces the cells should elongate along the axis where microtubules are aligned. This type of experiment has already been done by the authors to some extent when they laser-dissected the hinge from blade at 18hAPF and observed no further cell elongation. However, smaller cuts as designed in Bonnet et al 2012 (<https://royalsocietypublishing.org/doi/epdf/10.1098/rsif.2012.0263>) would be far more instructive.

Answer: We agree with the reviewer that this would be an elegant way to show microtubule pushing forces (though still not direct evidence). We performed this experiment (cell isolation from surrounding neighbors) previously (Singh et al., NCB), but we were not able to track changes for an extended period of time: The signal rapidly bleached out, and after a short time period (2 min), the tissue responded by wound healing (as cutting induces ROS release and Ca signaling). In addition, we had to ablate connections from other cells quite extensively (small micro-cuts are not working because cells stay connected to the neighboring cells extensively), making it hard to distinguish between different contributions.

- Modulate the stability/force generation of microtubules without altering their alignment. Whilst the authors attempted this with the pat RNAi unfortunately the alignment of microtubules was

also disrupted. If the authors could increase microtubule stability and show an increase in cell elongation as their model suggests, this would provide direct evidence to support their hypothesis.

Answer: We agree with the reviewer that the reversed experiment (i.e. increasing the stability of microtubules) should lead to longer cells. As non-centrosomal microtubules in wing cells are Patronin dependent, we overexpressed Patronin in these cells. Strikingly, we observed that cells are longer. These results were added to Figure 4h-l

Minor issues:

1) I have an issue with the measure of circularity, as a proxy for cell elongation. As circularity is a measure of how close to a perfect circle an objects shape is, a low value doesn't necessarily suggest that cells are elongated. Whilst I doubt this would change the results drastically, as it is doubtful any star shape cells exist, I would advise the authors based on the other shape descriptors they quantified (major and minor axis lengths) to use $1 - \text{minor axis}/\text{major axis}$ as a better read-out for cell elongation.

Answer: We thank the referee for pointing out this omission. We now added an analysis measuring major and minor axis. We find that during development cells concomitantly reduce the length of the minor axis as the length of the major axis increases. This data has been added to Figure 1.

2) Could the authors include a measurement of apical area to all perturbations and not only major and minor axis lengths, this would remove any ambiguity of whether apical area changes could explain differences in total wing length.

Answer: A measurement of the apical area has been added.

3) Could the authors either please include PD-axis guides for each image panel or highlight in the text that all images are aligned with PD-axis along the horizontal axis.

Answer: We thank the referee for this suggestion. PD-axis guides were added.

4) Could the authors please include an analysis in the WT examining the orientation of each cells major axis with respect to the PD axis, as highly variable orientation would put doubt into the contribution elongation makes to the overall tissue size.

Answer: We thank the reviewer for this great suggestion. Analysis of the orientation of the cell major axis with respect to the PD axis confirmed that cells elongated along the PD axis. This is in line with previous published data from the Eaton group, where they track the orientation of cell movements, oriented cell divisions, cell intercalation and cell elongation during pupal wing development. They show that during phase I (15-24 hAPF according to their definition), cells elongate along the PD direction (Aigouy et al., [https://www.cell.com/fulltext/S0092-8674\(10\)00890-1](https://www.cell.com/fulltext/S0092-8674(10)00890-1)).

REVIEWER COMMENTS

Reviewer #1 (Remarks to the Author):

The authors have significantly improved the manuscript by providing new quantifications, clarifications and new experimental data. The more extensive description of cell geometry is really useful and the new data on UAS-patronin is quite compelling.

I understood the experimental difficulty with nubbin Gal4 and MyoII depletion as well as the concern to avoid pleiotropic effects through long term depletion. I believe there is anyway enough convincing data to prove the point of the authors, and I am fully supportive for publication at this stage.

Maybe for the sake a clarity, it might be nice to add a couple of sentences in the legends of Figure 3 or in the main text to discuss the potential contribution of the posterior part of the tissue where contractility is stil high and the potential tug of war (otherwise readers might also be puzzled, like I was, by this very striking increase of the cell short axis upon MyoII depletion in the anterior compartment, which is hard to explain with the purely cell centric explanation provided at this stage).

Reviewer #2 (Remarks to the Author):

Most of the comments/questions were addressed with varying degree of thoroughness. A number of typos have snuck into the manuscript -- a few I noticed: Fig. 1h should read "principal cell axis", Fig. 2b and 2e should read "intensity", Fig. S1 should read "PD/AP axis length".

Reviewer #2 (Remarks on code availability):

The code is not accessible. Requires login and instructions are in German.

Reviewer #3 (Remarks to the Author):

I appreciate the effort the authors have gone to, to allay my concerns on the original manuscript submission.

- I appreciate the kymograph of E-CadGFP along cell junctions at high-temporal resolution which clearly shows that there is little-to-no junctional instability. I acknowledge that whilst this data clearly counteracts my alternative hypothesis, this result has no place within the manuscript as it would detract from the central message.

- The over-expression of patronin-GFP further elongating cells along the PD-axis is an important result and greatly improves the conclusions that microtubule dynamics drive cell elongation. However, I am surprised that this was not further extended to assess whether the wing also elongates in this genetic condition, in a similar analysis performed in Figure 5 for patronin RNAi and SqhGFP degradation conditions. I acknowledge that in my original review I did not mention extending this analysis beyond that of cells, so don't think publication should be beholden to this but it is something the authors might want to consider to further increase the impact of this work.

- I also acknowledge that the authors have toned down their statements on the protrusive force of microtubules, especially in the discussion. Furthermore, the over-expression of patronin-gfp shows that microtubule dynamics do induce cell elongation, further supporting the authors conclusions. Whilst I am happy with the manuscript, I still think the title is slightly too strong a statement and should be toned-down to reflect the more nuanced argument within the manuscript.

Based on the rebuttal letter and alterations to the manuscript, I have no concerns on the publication of this work. Whilst I do think the title is potentially too strong, I leave this judgement to the editor and still support publication if it is not changed. Furthermore, I have minor comments that I think will greatly improve the manuscript but leave these to the editor and authors discretion on whether they are acted upon:

1. The addition of cell area greatly enhances the manuscripts observations. However from Figure 1 and Supplementary Figure 1, cell area minimally decreases whilst cell height drastically decreases. Summed together this would suggest a volume decrease which the authors do not acknowledge. Whilst exploring this result is outside the scope of this work, as the manuscript starts with a detailed description on cell shape changes not commenting on this observation seems a major omission.

2. The authors have included cell area in several analyses of cell shape description; however, it is not consistently used throughout the manuscript (it is not present in Figure 3 for instance). As the inclusion of cell area in other experimental conditions makes the authors results more convincing, the inclusion of cell area throughout would also highlight that elongation is affected.

3. As the central message of the manuscript is on cell elongation driven by microtubule alignment, the use of circularity is still detracts from this. An example of this is in Figure 3g where the authors acknowledge that junctions become more circular thereby confounding the analysis performed in Figure 3d – is this increase in circularity driven by changes in elongation or the increased curvature of cell junctions? As the authors have data on the minor and major axis a measure of 1-width/length would provide a far more intuitive parameter to examine elongation than the current circularity.

We would like to thank the reviewers for their careful reading of the revised manuscript and valuable suggestions on improving it further. Following the referee's comments, we made additional changes to the manuscript (marked in blue).

REVIEWER COMMENTS

Reviewer #1 (Remarks to the Author):

The authors have significantly improved the manuscript by providing new quantifications, clarifications and new experimental data. The more extensive description of cell geometry is really useful, and the new data on UAS-patronin is quite compelling.

I understood the experimental difficulty with nubbin Gal4 and MyoII depletion as well as the concern to avoid pleiotropic effects through long term depletion. I believe there is anyway enough convincing data to prove the point of the authors, and I am fully supportive for publication at this stage.

Maybe for the sake of clarity, it might be nice to add a couple of sentences in the legends of Figure 3 or in the main text to discuss the potential contribution of the posterior part of the tissue where contractility is still high and the potential tug of war (otherwise readers might also be puzzled, like I was, by this very striking increase of the cell short axis upon MyoII depletion in the anterior compartment, which is hard to explain with the purely cell centric explanation provided at this stage).

Answer: We thank the referee for the positive feedback and support for publication. We appreciate the suggestion regarding a more thorough discussion of the effect of MyoII downregulation on cell shape changes, as this will indeed enhance the clarity of our findings and interpretation.

We concur with the reviewer regarding the significance of tissue mechanics in shaping cells and tissues. Here is how we envision that changes in single-cell mechanics propagate to the tissue level:

- At the single cell level, we reasoned that disrupted MyosinII activity should lead to a new equilibration of contractile stress, consequently resulting in cell rounding. Consistently, we found that the cells become rounder due to an increase in both length and even more by width. Our findings thus align with the concept that cells tend to become rounder when cortical tension and cellular pre-stress are released upon inhibition of MyosinII contractility (Mason et al., 2013; Royou et al., 2002; Rozbicki et al., 2015; Ochoa-Espinosa et al., 2017).
- At the tissue level, a reduction in myosin contractility may lead to tissue softening, thereby reducing resistance to microtubule pushing forces and further promoting elongation. As suggested by the reviewer, there may also be a contribution from the posterior compartment to the anterior. However, given the higher contractility in the posterior, one would anticipate a shorter anterior compartment.

Within this framework, the primary finding pertinent to our study is that cells do not become entirely spherical but instead elongate along the P/D axis. As we excluded the role of extrinsic

forces on cell shape at this stage, this suggests that microtubules are required (although not the only factor) for cell elongation within the plane.

Following the referee's recommendation, we made the following changes in the manuscript to further clarify this point:

- **Figure 1 legend:** Cartoon showing the effect of disruption of Myosin II and microtubule activity on cell shape. The cell shape changes result from a general release of cellular prestress upon loss of Myosin II contractility, as suggested by an increase in length and width of the cell. Thus, there is no direct relation between Myosin II polarized organization and cell elongation along the P/D axis. **Importantly, the reduction in Myosin II contractility may lead to tissue softening, which may also contribute to the observed cell shape changes.** However, Myosin II refines the overall cell shapes by (i) regulating the length of the cells and (ii) keeping the interfaces along the P/D axis taut and straight (indicated by black broken lines), as shown through the region of cell interfaces marked within the gray box.

Reviewer #2 (Remarks to the Author):

Most of the comments/questions were addressed with varying degree of thoroughness. A number of typos have snuck into the manuscript -- a few I noticed: Fig. 1h should read "principal cell axis", Fig. 2b and 2e should read "intensity", Fig. S1 should read "PD/AP axis length".

The code is not accessible. Requires login and instructions are in German.

Answer: We thank the reviewer for the positive feedback. We hope that our edits, as detailed below, will answer the remaining points:

- **Typos:** We thank the referee for pointing out this omission. We have carefully proofread the manuscript and corrected all typos accordingly.
- **Code availability:** We apologize for the inconvenience. We did not upload the script as the paper is not online yet. We guarantee that the code will be made accessible (incl. instructions in English) without requiring login credentials after acceptance of the paper.

Reviewer #3 (Remarks to the Author):

I appreciate the effort the authors have gone to, to allay my concerns on the original manuscript submission.

Answer: We are glad that the additional data and revisions have addressed your concerns.

- I appreciate the kymograph of E-CadGFP along cell junctions at high-temporal resolution which clearly shows that there is little-to-no junctional instability. I acknowledge that whilst this data clearly counteracts my alternative hypothesis, this result has no place within the manuscript as it would detract from the central message.

Answer: We acknowledge your point about the kymograph of E-CadGFP and its relevance to the central message. As suggested, we will not add it to the manuscript to maintain focus.

- The over-expression of patronin-GFP further elongating cells along the PD-axis is an important result and greatly improves the conclusions that microtubule dynamics drive cell elongation. However, I am surprised that this was not further extended to assess whether the wing also elongates in this genetic condition, in a similar analysis performed in Figure 5 for patronin RNAi and SqhGFP degradation conditions. I acknowledge that in my original review I did not mention extending this analysis beyond that of cells, so don't think publication should be beholden to this but it is something the authors might want to consider to further increase the impact of this work.

Answer: We appreciate your suggestion regarding extending the analysis of patronin-GFP overexpression to assess wing elongation. We have incorporated these additional data to demonstrate that the stabilization of microtubules is sufficient to drive cell elongation in an autonomous manner. However, while this manipulation effectively stabilizes non-centrosomal microtubules, it also disrupts centrosomal microtubules (spindle microtubules), resulting in cell division defects (see Fig. 1). Consequently, we cannot draw any conclusions regarding tissue shape.

- I also acknowledge that the authors have toned down their statements on the protrusive force of microtubules, especially in the discussion. Furthermore, the over-expression of patronin-gfp shows that microtubule dynamics do induce cell elongation, further supporting the authors conclusions. Whilst I am happy with the manuscript, I still think the title is slightly too strong a statement and should be toned-downed to reflect the more nuanced argument within the manuscript.

Answer: To remove any implicit suggestions of protrusive microtubule-based forces and to further strengthen the focus on the interplay between actin and MTs, we changed the title to '*Dynamic Interplay of Microtubule and Actomyosin Forces Drive Tissue Extension*'.

Based on the rebuttal letter and alterations to the manuscript, I have no concerns on the publication of this work. Whilst I do think the title is potentially too strong, I leave this judgement to the editor and still support publication if it is not changed. Furthermore, I have minor comments that I think will greatly improve the manuscript but leave these to the editor and authors discretion on whether they are acted upon:

Regarding the minor comments:

1. The addition of cell area greatly enhances the manuscripts observations. However from Figure 1 and Supplementary Figure 1, cell area minimally decreases whilst cell height drastically decreases. Summed together this would suggest a volume decrease which the authors do not acknowledge. Whilst exploring this result is outside the scope of this work, as the manuscript starts with a detailed description on cell shape changes not commenting on this observation seems a major omission.

Answer: We fully agree that an analysis of the 3D cell shape changes, including volume, would be very informative, but we are limited both experimentally and within the scope of the manuscript. Besides being very heterogeneous tissue, we couldn't obtain reliable staining with the basolateral marker Dlg-GFP (the signal showed the non-uniform distribution of endogenous Dlg-GFP), which precludes an accurate measurement of the 3D cell shape. The quantification of the basal side is especially tedious in this tissue, characterized by highly convoluted cellular morphology. As a result, we cannot accurately measure cell volume and decisively comment on it.

2. The authors have included cell area in several analyses of cell shape description; however, it is not consistently used throughout the manuscript (it is not present in Figure 3 for instance). As the inclusion of cell area in other experimental conditions makes the authors results more convincing, the inclusion of cell area throughout would also highlight that elongation is affected.

Answer: The analysis of the apical area in the MyosinII depletion experiments was omitted because previous studies have demonstrated that loss of MyosinII function leads to cortical relaxation, resulting in an expanded apical surface (Mason et al., 2013; Royou et al., 2002; Rozbicki et al., 2015; Ochoa-Espinosa et al., 2017). Instead, we focused on observing changes in cell shape. Interestingly, despite the release of pre-stress and the expected equilibration of forces, we did not observe complete cell rounding, as they are still elongated along the P/D axis. This suggests that in the absence of extrinsic forces, patterned microtubules are essential for cell elongation.

3. As the central message of the manuscript is on cell elongation driven by microtubule alignment, the use of circularity still detracts from this. An example of this is in Figure 3g where the authors acknowledge that junctions become more circular thereby confounding the analysis performed in Figure 3d – is this increase in circularity driven by changes in elongation or the increased curvature of cell junctions? As the authors have data on the minor and major axis a measure of 1-width/length would provide a far more intuitive parameter to examine elongation than the current circularity.

Answer: We appreciate this comment, and in general, we agree with the reviewer that the ratio of the minor and major axes (elongation index or aspect ratio) can be used as a measurement of cell elongation. However, since both axis lengths are provided, the ratio between them is just another presentation of the same data. Thus, we think that circularity is better suited to illustrate changes in shape (and it is dominated by the shape's gross features rather than the definition of its edges and corners).

REVIEWERS' COMMENTS

Reviewer #3 (Remarks to the Author):

I thank the authors for responding to my comments and providing more context to their experiments.

The title and conclusions the authors make are supported by the data, and I support publication of this work.